# On the Problem of Consistent Anomalies in Zero-Shot Industrial Anomaly Detection

**Tai Le-Gia**  *giataile@o.cnu.ac.kr*
*Department of Mathematics*
*Chungnam National University*

**Ahn Jaehyun**  *jhahn@cnu.ac.kr*
*Department of Mathematics*
*Chungnam National University*

**Reviewed on OpenReview:** *https: // openreview. net/ forum? id= o2MRb5QZ34*

## Abstract

Zero-shot image anomaly classification (AC) and anomaly segmentation (AS) play a crucial role in industrial quality control, where defects must be detected without prior training data. Current representation-based approaches rely on comparing patch features with nearest neighbors in unlabeled test images. However, these methods fail when faced with consistent anomalies—similar defects that consistently appear across multiple images—leading to poor AC/AS performance. We present Consistent-Anomaly Detection Graph (CoDeGraph), a novel algorithm that addresses this challenge by identifying and filtering consistent anomalies from similarity computations. Our key insight is that for industrial images, normal patches exhibit stable, gradually increasing similarity to other test images, whereas consistent-anomaly patches show abrupt spikes after exhausting a limited set of images with similar matches. We term this phenomenon "neighbor-burnout" and engineer a robust system to exploit it. CoDeGraph constructs an image-level graph, with images as nodes and edges linking those with shared consistent-anomaly patterns, using community detection to identify and filter out consistent-anomaly patches. To provide a theoretical explanation for this phenomenon, we develop a model grounded in Extreme Value Theory that explains why our approach is effective. Experimental results on MVTec AD using the ViT-L-14-336 backbone show 98.3% AUROC for AC and AS performance of 66.8% (+4.2%) F1 and 68.1% (+5.4%) AP over state-of-the-art zero-shot methods. Additional experiments with the DINOv2 backbone further enhance segmentation, achieving a 69.1% (+6.5%) F1 and a 71.9% (+9.2%) AP, demonstrating the robustness of our approach across different architectures. Our code is available at `https://github.com/DumBringer/CoDeGraph`.

## 1 Introduction

Anomaly detection is vital for industrial manufacturing, where products are semantically identical but defects range from subtle scratches to logical and structural errors. Traditional full-shot methods (Batzner et al., 2024; Defard et al., 2021; Mousakhan et al., 2024; Li et al., 2021; Rudolph et al., 2023; Roth et al., 2022; Zhou et al., 2024) deliver strong AC/AS performance by relying on a large corpus of normal training images. However, the fast-paced and diverse nature of industrial settings often demands solutions that require minimal resources. Few-shot approaches, such as RegAD (Huang et al., 2022) and GraphCore (Xie et al., 2023), have shown reliable accuracy by utilizing a small handful of normal images. Zero-shot methods surpass this limit by completely removing the requirement for training data. Techniques like WinCLIP (Jeong et al., 2023), AnomalyGPT (Gu et al., 2024), AnomalyCLIP (Zhou et al., 2023), and APRIL-GAN (Chen et al., 2023) pioneer the use of text prompts to guide anomaly detection. More recently, MuSc (Li et al., 2024)

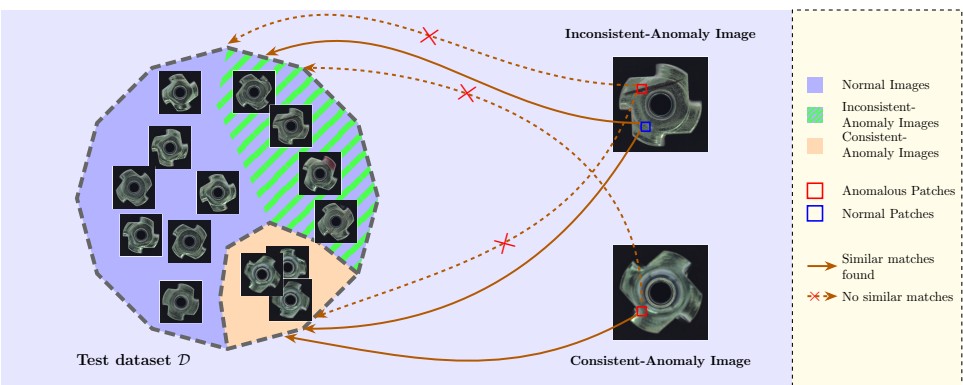

Figure 1: Illustration of zero-shot anomaly detection's consistent-anomaly problem. Industrial images have normal patches (blue squares) that match nearly all test images. Scratches and other random anomalies have high anomaly scores since they fail to find similar matches across the test set. Defects from consistent-anomaly images (flipped metal nuts) easily find deceptive matches within the images (orange region) sharing the same anomaly pattern (rotate counter-clockwise instead of clockwise).

introduces a mutual scoring mechanism that compares patches across unlabeled test images, offering a robust zero-shot solution.

Despite these advances, existing zero-shot methods face a critical limitation with *consistent anomalies*—similar defects that recur across multiple images in a test dataset. For instance, in MVTec AD Bergmann et al. (2019), anomalies like flipped `metal_nut` objects form isolated groups with distinct features, as shown in Fig. 1. They are distinguishable at the image level but problematic for patch-level or text-prompt approaches (Jeong et al., 2023), resulting in false negatives. This motivates methods exploiting image-level relationships for zero-shot industrial anomaly detection.

To address this challenge, we propose CoDeGraph (Consistent-Anomaly Detection Graph), a novel approach that leverages image-level relationships rather than purely patch-level comparisons. Our key insight is that consistent anomalies exhibit a distinctive *neighbor-burnout* phenomenon: normal patches retain steady similarity to neighboring images due to abundant similar matches across images, characterized by a power-law decay rate, whereas consistent-anomaly patches show stable distances to a limited set of similar anomalous images followed by abrupt distance spikes when these matches are exhausted.

CoDeGraph operationalizes this observation through a three-stage framework. First, by introducing the *endurance ratio* that captures the neighbor-burnout phenomenon, we enable the identification of connections that potentially connect similar anomalous patches across images. Second, we construct a similarity graph where nodes represent test images and edge weights reflect the strength of suspicious connections between them, enabling consistent-anomaly images to form densely connected communities. Finally, we apply community detection algorithms to identify these dense communities and selectively filter patches within them that exhibit strong dependency on intra-community matches, thereby removing the source of scoring bias while preserving useful normal patches for the comparison.

This graph-based approach enables CoDeGraph to robustly identify and mitigate the impact of consistent anomalies while maintaining robust performance on other datasets. Experimental results demonstrate that CoDeGraph achieves competitive performance on inconsistent-anomaly datasets, including the inconsistent subclasses of MVTec AD (98.3% AUROC-cls, 65.1% F1-seg) and Visa (91.6% AUROC-cls, 48.3% F1-seg), while simultaneously providing substantial improvements on consistent-anomaly datasets with gains of up to 14.9% in F1-score and 18.8% in AP for segmentation tasks. The main contributions of our work are

- We formalize the consistent-anomaly problem and identify the neighbor-burnout phenomenon, developing an Extreme Value Theory model explaining why normal patches follow predictable power-law decay while consistent anomalies deviate sharply. Motivated by this insight, we propose the endurance ratio metric for consistent-anomaly discrimination.

- We propose CoDeGraph, a novel framework that constructs an image-level similarity graph based on our endurance ratio. This approach enables the robust identification of consistent-anomaly images as dense communities, which are then selectively filtered to mitigate scoring bias from deceptive matches.

- Our CodeGraph achieves state-of-the-art performance on consistent-anomaly benchmarks (gains up to 14.9% F1-score, 18.8% AP) while maintaining competitive performance on conventional datasets, proving effectiveness as a general zero-shot solution.

## 2 Preliminary

In zero-shot anomaly detection and segmentation, we aim to identify defects in unlabeled test images $\mathcal{D} = \{I_1, \ldots, I_N\}$ without any training data. Our method builds upon two key components from existing work: Local Neighborhood Aggregation and Mutual Scoring Mechanism (Li et al., 2024).

### 2.1 Local Neighborhood Aggregation (LNAMD)

LNAMD (Li et al., 2024) processes Vision Transformer (ViT) (Dosovitskiy et al., 2021) features at multiple receptive fields to capture anomalies of varying sizes. For patch tokens at ViT layer $l$ of image $I_i$, $F_i^l = \left[x_{i,1}^l, \ldots, x_{i,M}^l\right] \in \mathbb{R}^{M \times C}$ where $M$ is the number of patches and $C$ is the feature dimension, the process begins by reshaping patch tokens $F_i^l$ to $\sqrt{M} \times \sqrt{M} \times C$. An adaptive pooling operation is then applied in $r \times r$ neighborhoods at each spatial location to capture multi-scale features. The output aggregated patch tokens are reshaped back to size $M \times C$, resulting in $F_i^{l,r} = \left[p^r(x_{i,1}^l), \ldots, p^r(x_{i,M}^l)\right]$ where $p^r$ is an adaptive pooling operator of size $r$.

### 2.2 Mutual Scoring Mechanism

Mutual Scoring Mechanism (MSM) (Li et al., 2024) computes anomaly scores by comparing each patch in a query $\mathcal{Q}$ to patches of all images in the base $\mathcal{B}$. In zero-shot settings, the query set and the base set are simply identical to the test dataset $\mathcal{D} = \{I_1, \ldots, I_N\}$. Given a patch $x_{i,m}^l$ in image $I_i \in \mathcal{Q}$, its distance to an image $I_j \in \mathcal{B}$[1] at layer $l$ and receptive field $r$ is:

$$d^r(x_{i,m}^l, I_j) = \min_n \|p^r(x_{i,m}^l) - p^r(x_{j,n}^l)\|_2^2.$$

The distances to all other images in $\mathcal{B}$ are collected into a vector and sorted:

$$D_{\mathcal{B}}^r(x_{i,m}^l) = \left[d^r(x_{i,m}^l, I_{(1)}), \ldots, d^r(x_{i,m}^l, I_{(N-1)})\right],$$

where $d^r(x_{i,m}^l, I_{(i)})$ is the i-th smallest distance from $x_{i,m}^l$ to the images in the base set $\mathcal{B}$. We refer to this process as *Mutual Similarity Ranking* (MSR), and the resulting vector as *mutual similarity vector*. For more stable scoring, Li et al. (2024) proposes applying an interval average operation on the $K$ smallest elements:

$$a_{\mathcal{B}}^r(x_{i,m}^l) = \frac{1}{K} \sum_{k=1}^{K} d^r(x_{i,m}^l, I_{(k)}). \tag{1}$$

The final anomaly score for patch $m$ is computed as the expectation over all selected layers and receptive fields:

$$\mathcal{A}_{\mathcal{B}}(x_{i,m}) = \mathbb{E}_{l,r}\left[a_{\mathcal{B}}^r(x_{i,m}^l)\right]. \tag{2}$$

---

[1]Throughout this paper, when we refer to the similarity (or distance) of a patch to an image, we mean the similarity (or distance) between the patch and the most similar patch found within that image. We use the terms "similarity of a patch to an image" and "distance of a patch to an image" interchangeably, where distance represents the inverse of similarity.

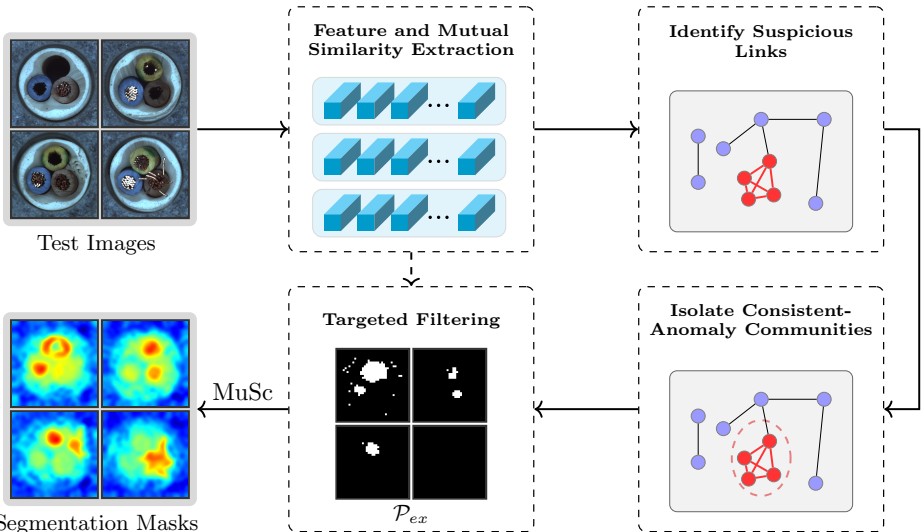

Figure 2: Overview of *CoDeGraph*.

# 3 Proposed Method

## 3.1 Overview

CoDeGraph addresses consistent anomalies through a three-stage pipeline: identify, isolate, and filter. Subsection 3.2 introduces the neighbor-burnout phenomenon and the endurance ratio metric for quantifying it, thus enabling detection of connections between consistent-anomaly patches. Subsection 3.3 presents the construction of an image-level similarity graph, where nodes represent images and edges encode shared consistent anomalies, enabling consistent-anomaly images to form densely connected communities. Subsection 3.4 details the community detection methodology and targeted filtering strategy, which identifies consistent-anomaly images as outlier communities and selectively removes anomalous patches from the base set $\mathcal{B}$ while preserving useful normal patches.

Throughout this section, we set the receptive field parameter $r = 1$ unless specified. We use $r > 1$ only for the final anomaly score calculations.

## 3.2 Mutual Similarity Analysis

Let's first formalize the concept of consistent and consistent-anomaly patches.

**Definition 3.1** ($\epsilon$-consistent)**.** Given a metric distance $d$, a patch token $x$ from image $I$ is called $\epsilon$-consistent if there exists at least one patch token $x'$ in some image $I' \neq I$ such that $d(x, x') < \epsilon$. We say there exists an $\epsilon$-consistent link from $x$ to $I'$ in this case, and the set of all such images $I'$ is called the $\epsilon$-consistent neighbors of $x$.

**Definition 3.2** ($\epsilon$-consistent-anomaly)**.** An $\epsilon$-consistent link is called an $\epsilon$-consistent-anomaly link if the ground truth label of the source patch is anomalous. Similarly, an $\epsilon$-consistent patch is called an $\epsilon$-consistent-anomaly patch if its ground truth label is anomalous.

For an anomalous patch $x_a$ with an $\epsilon$-consistent neighbors of size $H$ in $\mathcal{B}$, we can derive the bound,

$$K \cdot a_{\mathcal{B}}^r(x_a) \leq H_0 \epsilon + \sum_{i=H_0+1}^{K} d^r(x_a, I_{(i)}),$$

where $H_0 = \min(H, K)$. When $\epsilon$ is sufficiently small (less than typical distances between similar normal patches) and the ratio $H_0/K$ is substantial, $a_{\mathcal{B}}^r(x_a)$ fails to distinguish $\epsilon$-consistent-anomaly patches from

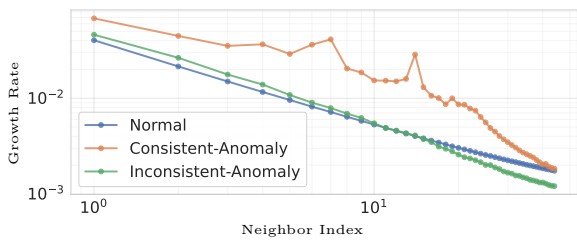

(a) `Cable`: Avg. Growth Rate

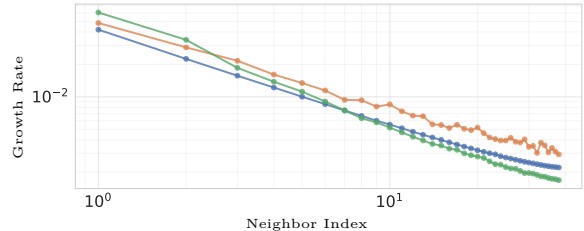

(b) `Capsule`: Avg. Growth Rate

Figure 3: Log-log plots of avg. growth rate $\tau^{(i)}(x)$. While normal patches and inconsistent anomalies in the `Cable` of MVTec AD show power-law decay in the growth rate, consistent anomalies show neighbor-burnout with a sudden rise (orange) in $\tau^{(i)}(x)$ after exhausting similar matches. All patches in `Capsule`, which has a minimum presence of consistent anomalies, exhibit power-law decay in the growth.

normal patches. Such patches usually appear in scenarios with repeated similar anomalies, such as logical flaws (flipped `metal_nuts`, contaminated `pills`) or structural flaws (missing components). This behavior contrasts with random defects like scratches or cracks, which exhibit unique patterns. Our goal is to isolate these consistent anomalies from the base set $\mathcal{B}$.

Throughout the rest of the paper, unless otherwise specified, consistent-anomaly patches refers to $\epsilon$-consistent-anomaly patches with sufficiently small $\epsilon$. For consistent-anomaly benchmarks, we define them as anomalous patches whose anomaly scores fall below the 80th percentile of normal patch scores, indicating deceptive matches.

**Neighbor-burnout phenomenon.** Patch distances to their neighbors evolve differently for normal and consistent-anomaly patches. Let $d(x, I_{(i)})$ denote the distance to the $i$-th nearest image in $\mathcal{B}$, we define the (log) similarity growth rate $\tau$ at index $i$ as follows:

$$\tau^{(i)}(x) = \ln\left(\frac{d(x, I_{(i+1)})}{d(x, I_{(i)})}\right). \tag{3}$$

For normal patches, mutual similarity vectors grow gradually due to abundant similar matches in the base set $\mathcal{B}$, resulting in small, stable values of $\tau^{(i)}(x)$. Empirically, we observed power-law decay in the growth of $\mathbb{E}[\tau^{(i)}(x)]$ and $\mathrm{Var}[\tau^{(i)}(x)]$, as demonstrated in Fig. 3. To provide a formal basis for this empirical finding, we developed a theoretical model grounded in Extreme Value Theory. Our model rests on two key assumptions: that patch-to-patch similarity distributions have a power-law tail (validated in Appendix Fig.8), and that distances can be approximated as i.i.d. random variables. While we acknowledge spatial correlations exist, the large number of patches per image (e.g., 1369) justifies the i.i.d. approximation and thus makes our model mathematically tractable. These assumptions, along with the Fisher-Tippet-Gnedenko theorem (Fisher & Tippett, 1928) and results from ordered statistics, lead to the following result:

**Theorem 3.1** (Similarity Growth Dynamics). *Under the assumptions of our model, the similarity growth rate $\tau^{(i)}(x)$ for a normal patch $x$ at neighbor index $i$ is exponentially distributed:*

$$\tau^{(i)}(x) \sim \mathrm{Exp}(\alpha \cdot i),$$

*where $\alpha$ is the tail index of the underlying similarity distribution. Consequently, the expectation and variance decay with the neighbor index $i$ are*

$$\mathbb{E}[\tau^{(i)}(x)] = \frac{1}{\alpha i}, \quad \mathrm{Var}[\tau^{(i)}(x)] = \frac{1}{(\alpha i)^2}.$$

*The full derivation and formal statement are provided in Appendix A.1.*

This theorem establishes a predictable statistical baseline for normal patches, characterized by stable, power-law decay in growth rates. In contrast, consistent-anomaly patches $x_a$ exhibit a fundamentally different

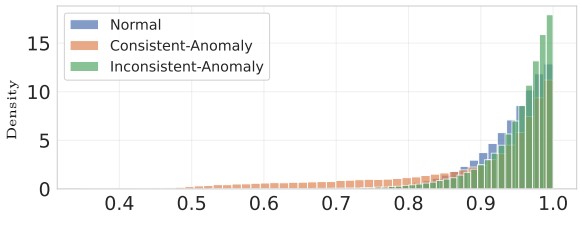
(a) Distribution of endurance ratios $\zeta(x, I_{(i)})$

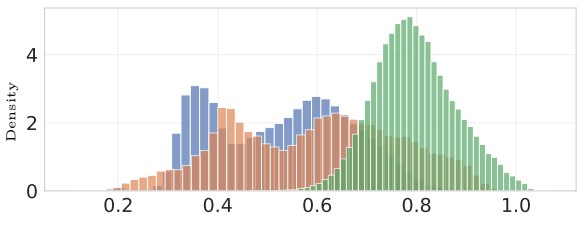
(b) Distribution of absolute distances $d(x, I_{(i)})$

Figure 4: Distributions of $\zeta(x, I_{(i)})$ and $d(x, I_{(i)})$ for $i < \omega$ across patch types on `cable`. (a) The endurance ratio provides clear domination at the tail, with consistent-anomaly patches exhibiting significantly lower $\zeta$, enabling robust identification of suspicious links. (b) Absolute distances $d(x, I)$ show overlapping distributions between normal patches and consistent anomalies, making discrimination challenging.

behavior we term *neighbor-burnout*: their distances to the images in $\mathcal{B}$ spike abruptly after exhausting a limited set of $H$ $\epsilon$-consistent neighbors. This manifests as $\tau^{(i)}(x_a)$ remaining small for $i < H$, but then jumping significantly for $i \geq H$, creating a distinct discontinuity in the mutual similarity vector. The neighbor-burnout phenomenon thus represents a statistically significant deviation from the expected growth rate dynamics at $\tau^{(H)}(x)$. We leverage this phenomenon by defining the endurance ratio:

$$\zeta(x, I_{(i)}) = \frac{d(x, I_{(i)})}{d(x, I_{(\omega)})}, \tag{4}$$

where $\omega > H$ serves as a reference index beyond the burnout point. This ratio becomes exceptionally small for consistent-anomaly patches, as early distances remain small while later distances grow large after exhausting similar matches. Normal patches typically exhibit more moderate ratios due to the power law decay in mutual similarity growth. In practice, setting $\omega$ sufficiently large (e.g., $\omega = 0.5 \cdot N$) effectively captures this disparity and provides a robust discriminative signal for identifying consistent-anomaly links.

**Suspicious links**. Using the endurance ratio, we define our set of suspicious links as the set of links with the smallest endurance ratios:

$$\mathcal{S}_l = \left\{ (x, I_{(i)}) \in L \mid \zeta(x, I_{(i)}) \leq \lambda \right\},$$

where $L = \{(x, I) \mid I \text{ is an image in } \mathcal{B}, \text{ x is a patch} \notin I\}$ is the set of all possible links, and $\lambda$ is a threshold controlling the number of links in $\mathcal{S}_l$. This set $\mathcal{S}_l$ forms the foundation of our graph-based approach by capturing links likely to be consistent-anomaly links. While this basic formulation provides a good starting point, zero-shot settings require additional robustness, which we address in the next subsection.

### 3.3 Anomaly Similarity Graph

We construct an image-level similarity graph $\mathcal{G} = (V, E)$ to model relationships among images in base set $\mathcal{B}$ and to identify shared anomalous patterns. The graph's basic construction comprises the following: each node $v \in V$ represents an image in $\mathcal{B}$, an edge $(I_i, I_j) \in E$ exists if at least one suspicious link in $\mathcal{S}_l$ connects a patch between $I_i$ and $I_j$, and the weight $w_{ij}$ counts the number of suspicious links between $I_i$ and $I_j$.

The graph is designed to be discriminative: images with consistent anomalies form densely connected communities with high edge weights, distinct from normal images or those with random anomalies. This is achievable because consistent anomalies yield strong, systematic connections due to a large number of low-$\zeta$ links concentrated within a small subset of consistent-anomaly images, unlike the sparser links of normal or random-anomaly patches, as visualized in Fig. 5. In zero-shot settings, robustly achieving this discriminativeness is challenging. Hence, to enhance the graph's ability to isolate consistent-anomaly communities, we introduce two key refinements: the *Weighted Endurance Ratio* and *Coverage-based Link Selection*.

**Weighted endurance ratio.** Although the links from normal patches are often widely distributed across the graph $\mathcal{G}$, their overpresence can blur the distinctiveness of consistent-anomaly communities. This issue becomes clear in datasets with high normal pattern variability, such as `breakfast_box` in MVTec LOCO. Such variability leads to more normal patches exhibiting high and unstable growth rates $\tau^{(i)}$, resulting in an

increased number of normal links with small endurance ratios $\zeta$. To address this, we introduce the weighted endurance ratio:

$$\zeta'(x, I_{(i)}) = \zeta(x, I_{(i)}) \cdot d(x, I_{(i)})^{-\alpha} = \frac{d(x, I_{(i)})^{1-\alpha}}{d(x, I_{(\omega)})}.$$

By incorporating the inverse distance $d\left(x, I_{(i)}\right)^{-\alpha}$, this formulation prioritizes links with larger absolute distances, typically associated with anomalous patches, including consistent-anomaly patches. Thereby, $\zeta'$ amplifies the presence of anomalous patches in $\mathcal{S}_l$ while suppressing normal links, as shown in Fig. 7. Additionally, in MSM 2.2, removing normal patches from the base set $\mathcal{B}$ is more harmful than filtering anomalous ones. The weighted endurance ratio biases filtration toward anomalous patches, preserving normal patches in $\mathcal{B}$ and hence maintaining the core strength of MSM.

**Coverage-based Selection**. For $\mathcal{S}_l$ construction, we introduce a coverage-based approach. The core concept is simple: gradually increase $\lambda$ until the resulting graph achieves sufficient coverage—defined as the percentage of nodes $v$ with $d(v) > 0$. Such an approach ensures the graph captures enough connections in the background nodes to make the consistent-anomaly communities distinctive without being overwhelmed by normal links. In our implementation, we achieve this efficiently through an incremental link addition process, captured in Algorithm 1. The target coverage $\tau$ was set to 0.95 to ensure that most images were represented in the graph while allowing for potential isolation of outlier nodes, thereby avoiding the situation of adding an infinite number of links. Further details regarding the stopping time of Algorithm 1 are given in Appendix A.2.

---

**Algorithm 1** Coverage-based Selection

1: **function** COVERAGESELECTION($L, \tau, N$)
2:      Sort $L$ by $\zeta'$ ascending
3:      $\mathcal{S}_l \leftarrow \emptyset$
4:      $k \leftarrow N(N-1)/2$    ▷ Initial number of links
5:      **repeat**
6:          Add top-$k$ links from $L$ to $\mathcal{S}_l$
7:          $c \leftarrow$ fraction of nodes with degree $\geq 1$
8:          **if** $c < \tau$ **then** $k \leftarrow k + N(N-1)/2$
9:      **until** $c \geq \tau$ or all links used
10:      **return** $\mathcal{S}_l$

**Algorithm 2** Targeted Patch Filtering

1: **function** TARGETEDFILTERING($\mathcal{S}_c, \mathcal{B}$)
2:      $\mathcal{P}_{\text{ex}} \leftarrow \emptyset$; Compute $a_{\mathcal{B}}(p)$ for all patches $p$
3:      **for** each $C_i \in \mathcal{S}_c$ **do**
4:          $\mathcal{B}_{\text{temp}} \leftarrow \mathcal{B} \setminus C_i$; $R \leftarrow []$
5:          **for** each patch $p$ **do**
6:             $r(p) \leftarrow a_{\mathcal{B}_{\text{temp}}}(p)/a_{\mathcal{B}}(p)$
7:             Append $r(p)$ to $R$
8:          $\theta \leftarrow$ 99th percentile of $\{r(p)|p \in \mathcal{B} \setminus C_i\}$
9:          $\mathcal{P}_{\text{ex}} \leftarrow \mathcal{P}_{\text{ex}} \cup \{p \in C_i|r(p) > \theta\}$
10:      **return** $\mathcal{P}_{\text{ex}}$

---

### 3.4 Community Detection and Filtering

We employ the Leiden algorithm (Traag et al., 2019) with the Constant Potts Model (CPM) (Traag et al., 2011) to identify communities sharing consistent anomalies. The CPM optimizes the following objective function:

$$Q = \sum_{ij}(A_{ij} - \gamma)\delta(\sigma_i, \sigma_j),$$

where $A_{ij}$ is the adjacency matrix and $\delta(\sigma_i, \sigma_j) = 1$ if nodes $i$ and $j$ belong to the same community. The resolution parameter $\gamma$ ensures communities have density exceeding $\gamma$ while inter-community density remains below $\gamma$ (Traag et al., 2011). Thus, we set $\gamma$ to the 25th quantile of edge weights to ensure that communities with sufficient density were identified. Community density is $\rho(C) = \sum_{u,v \in C} w_{uv}/n_C(n_C - 1)$ where $n_C$ is the number of nodes in community $C$. We choose CPM over popular modularity-based methods (Newman & Girvan, 2004) as the latter fragment consistent-anomaly communities due to degree-based null model assumptions. Detailed analysis of the community detection algorithms is presented in Appendix B.1.

Let $\mathcal{C} = \{C_1, ..., C_h\}$ be detected communities. To identify those containing consistent anomalies (exhibiting exceptionally high density), we apply Tukey's fences IQR outlier detection Tukey et al. (1977); Carling (2000) to community densities $\{\rho(C_i)|n_{C_i} > 1\}_1^h$. Communities with density exceeding $Q_3 + k_{\text{IQR}} \cdot IQR$ are flagged as outliers, where we set $k_{\text{IQR}} = 4.5$ to ensure only profoundly connected communities are identified. We denote the set of all outlier communities as $\mathcal{S}_c$.

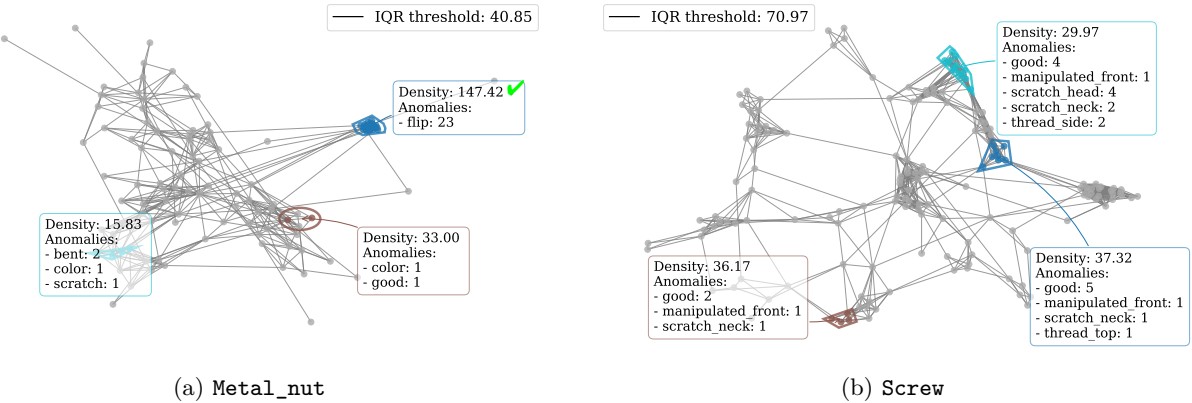

(a) `Metal_nut`                    (b) `Screw`

Figure 5: Anomaly similarity graphs on MVTec AD subclasses showing top three communities by density. (a) `Metal_Nut`: Community #1 contains all 23 flipped metal nuts with exceptionally high density, exceeding the IQR threshold. (b) `Screw`: Nodes are clustered into distinct communities, but none exhibit exceptionally high density.

**Targeted Patch Filtering.** Rather than removing entire $\mathcal{S}c$ from $\mathcal{B}$, we selectively target patches with strong intra-community dependency. Consistent-anomaly patches rely heavily on matches within their community $C_i$, causing significant score increases when these matches are excluded, unlike normal patches that find abundant external matches outside $C_i$. Concretely, for each $C_i \in \mathcal{S}c$, we compute score ratio $r(p) = a_{\mathcal{B}\setminus C_i}(p)/a_{\mathcal{B}}(p)$, where high ratios indicate strong dependency. Our method adaptively sets threshold $\theta$ to the 99th percentile of ratios from patches outside $C_i$, automatically identifying exceptionally dependent patches without manual tuning. The filtering process is formalized in Algorithm 2.

### 3.5   Final Anomaly Scores

The final anomaly scores are computed using the original Mutual Scoring Mechanism from Section 2.2 using multiple receptive fields $r$, but with the new base $\mathcal{B}_{\text{final}} = \mathcal{B} \setminus \mathcal{P}_{\text{ex}}$, where $\mathcal{P}_{\text{ex}}$ is the set of patches to be excluded, resulting from Algorithm 2. This ensures that anomaly scores reflect true anomalies without being biased by deceptively close matches from consistent anomalous patterns. This targeted approach preserves useful information from normal patches within outlier communities while effectively addressing the scoring bias introduced by consistent anomalies.

## 4   Experiments

We evaluated CoDeGraph's performance for zero-shot AC and AS on standard and newly introduced benchmarks to evaluate its ability to detect consistent anomalies while maintaining robust performance on datasets without consistent anomalies. The experiments demonstrate that CoDeGraph outperforms SOTA zero-shot methods on consistent-anomaly datasets and delivers competitive results on inconsistent-anomaly datasets.

### 4.1   Experimental Setup

**Datasets.** We conducted experiments on two well-established benchmarks for industrial AC and AS: MVTec AD (Bergmann et al., 2019) and Visa (Zou et al., 2022). MVTec AD (15 classes) is divided into MVTec-**C**onsistent**A**nomaly (`cable`, `metal_nut`, `pill`) exhibiting strong consistent-anomaly presence and MVTec-**I**nconsistent**A**nomaly (remaining 12 classes). Visa provides diverse, subtle defects for inconsistent-anomaly evaluation. To address the lack of benchmarks tailored for consistent-anomaly evaluation, we introduced two novel benchmarks: *MVTec-SynCA* and *ConsistAD*. MVTec-SynCA applies subtle transformations (lighting changes, camera shifts) to a single anomalous image for each subclass in MVTec AD, simulating consistent anomalies. ConsistAD comprises consistent-anomaly subclasses from MVTec AD, MVTec LOCO (Bergmann

Table 1: Quantitative comparisons on **Consistent-Anomaly Datasets**. We compared CoDeGraph with state-of-the-art zero-shot methods. Bold indicates the best performance. All metrics are in %.

| Dataset | Method | AUROC-cls | F1-cls | AP-cls | AUROC-seg | F1-seg | AP-seg | PRO-seg |
|---|---|---|---|---|---|---|---|---|
| MVTec-CA | AnomalyCLIP | 81.3 | 87.7 | 91.7 | 81.8 | 29.2 | 24.3 | 64.4 |
| | WinCLIP | 87.6 | 90.8 | 95.4 | 72.6 | 23.2 | - | 46.6 |
| | APRIL-GAN | 79.1 | 88.5 | 93.7 | 71.3 | 26.6 | 22.6 | 43.2 |
| | ACR | 73.3 | 88.1 | 88.1 | 86.9 | 42.7 | 35.5 | 66.0 |
| | MuSc 30% | 97.2 | 97.0 | 99.3 | 93.3 | 58.9 | 58.4 | 92.8 |
| | MuSc 10% | 94.1 | 94.9 | 98.7 | 88.3 | 51.6 | 51.2 | 90.1 |
| | CoDeGraph (Ours) | **98.5** (↑ 1.3) | **97.8**(↑ 0.8) | **99.6**(↑ 0.3) | **98.1**(↑ 4.8) | **73.8**(↑ 14.9) | **77.2**(↑ 18.8) | **95.4**(↑ 2.6) |
| MVTec-SynCA | AnomalyCLIP | 87.4 | 92.4 | 95.7 | 89.9 | 37.9 | 33.5 | 77.6 |
| | APRIL-GAN | 81.3 | 90.2 | 92.4 | 85.7 | 42.1 | 39.7 | 41.0 |
| | WinCLIP | 89.9 | 93.8 | 96.5 | 81.5 | 26.0 | 18.7 | 59.2 |
| | MuSc 10% | 88.8 | 92.3 | 96.8 | 90.7 | 50.8 | 48.3 | 82.9 |
| | CoDeGraph (Ours) | **96.8**(↑ 6.9) | **97.2**(↑ 3.4) | **99.0**(↑ 2.2) | **97.2**(↑ 6.5) | **63.2**(↑ 12.4) | **63.3**(↑ 15.0) | **91.1**(↑ 8.2) |
| ConsistAD | AnomalyCLIP | 75.1 | 73.8 | 76.3 | 73.6 | 31.1 | 25.2 | 53.2 |
| | WinCLIP | 76.6 | 74.5 | 75.9 | 60.8 | 24.7 | 18.8 | 41.8 |
| | APRIL-GAN | 68.7 | 73.9 | 69.2 | 61.1 | 24.3 | 19.9 | 21.7 |
| | MuSc 10% | 88.9 | 84.8 | 88.9 | 81.7 | 44.6 | 43.6 | 78.9 |
| | CoDeGraph (Ours) | **91.0**(↑ 2.1) | **87.9**(↑ 3.1) | **90.3**(↑ 1.4) | **86.9**(↑ 5.2) | **55.9**(↑ 11.3) | **57.5**(↑ 13.9) | **82.5**(↑ 3.6) |

et al., 2022), and MANTA (Fan et al., 2024). Details on MVTec-SynCA and ConsistAD are provided in Appendix C.

**Implementation Details.** All experiments use fixed parameters (described below) across all datasets, unless specified in ablations. For fair comparison with baselines, we primarily used ViT-L/14-336, pre-trained by OpenAI Radford et al. (2021), as the feature extraction backbone. This model consists of 24 layers, organized into four stages of six layers each. Patch tokens were extracted from layers 6, 12, 18, and 24, following Chen et al. (2023); Li et al. (2024). The linearly projected class token from the final layer was employed for classification optimization with RsCIN Li et al. (2024). All unlabeled test images were resized to $518 \times 518$ pixels. While we report CLIP ViT-L/14-336 results for fair comparison with existing methods, experiments with different architectures such as DINO(Caron et al., 2021) and DINOv2 (Oquab et al., 2023) demonstrated that DINOv2-L-14 achieved superior segmentation performance (89.9% pixel-wise AUROC, 69.1% pixel-wise F1, 71.9% pixel-wise AP). Detailed results on different ViT structures are in Appendix B.6.

For the anomaly similarity graph, we selected the distance $d(x, I_{(i)})$ to the i-th nearest image, as used in equation 3 and equation 4, as the average of distances to the i-th nearest image across the selected layers:

$$d_{\text{agg}}(x, I_{(i)}) = \frac{1}{4} \sum_{l \in \{6,12,18,24\}} d^{r=1}(x^l, I_{(i)}).$$

This averaging strategy enabled efficient graph construction and captured multi-level semantic relationships between patches. Consequently, each patch-image pair in $S_l$ established four links, corresponding to patch tokens from the four selected layers, rather than a single link. The weighted endurance ratio was set with $\alpha = 0.2$ and $\omega = 0.3 \cdot N$. The coverage-based selection algorithm targeted a coverage of $\tau = 0.95$. For anomaly scoring via the MSM in equation 2, we averaged the lowest 10% of distances instead of the 30% used in prior work (Li et al., 2024) (for the discussion on this decision, see Appendix B.7). For the final anomaly scores, we used receptive field sizes $r \in \{1, 3, 5\}$ to enhance AS of both small and large defects. All main experiments use fixed hyperparameters across all datasets, with no manual per-dataset tuning except ablation study.

**Evaluation Metrics.** For AC, we reported three metrics: Area Under the Receiver Operating Characteristic curve (AUROC), Average Precision (AP), and F1-score at the optimal threshold (F1). For AS, we reported four metrics: pixel-wise AUROC, pixel-wise F1, pixel-wise AP, and Area Under the Per-Region Overlap Curve (AUPRO). For the evaluation metrics related to consistent anomalies, we selected consistent anomalies as anomalous patches with anomaly scores in the full base $\mathcal{B}$ (i.e., MuSc) below the 80th percentile of normal patch scores. Other anomalies were labeled as inconsistent-anomaly patches. This threshold uses ground-truth solely for post-hoc analysis and evaluation metrics, ensuring no impact on the zero-shot nature of CoDeGraph.

**Baselines.** We compared CoDeGraph against SOTA zero-shot methods, including WinCLIP (Jeong et al., 2023), APRIL-GAN (Chen et al., 2023), AnomalyCLIP (Zhou et al., 2023), ACR (Li et al., 2023), and MuSc Li et al. (2024). For MuSc, we reported two versions: MuSc 30% (reported in Li et al. (2024)) and MuSc 10%, where the percentage indicates the size of the interval average operation from equation 1. Additionally, we compare with representative few-shot methods and full-shot methods in Appendix B.8. When baseline metrics were unavailable, we reproduced results using official implementations.

## 4.2 Quantitative and qualitative results

**Consistent-Anomaly Datasets.** The results for consistent-anomaly datasets are summarized in Table 1. On the MVTec-CA dataset, CoDeGraph recorded an image-level AUROC of 98.5%, which was 1.3% higher than MuSc and comparable to state-of-the-art full-shot methods [2]. For segmentation metrics, CoDeGraph showed a 14.9% increase in F1 and an 18.8% increase in AP compared to the next best zero-shot method. These results highlight challenges faced by existing zero-shot methods in addressing consistent anomalies, especially for segmentation tasks.

Experiments on the MVTec-SynCA and ConsistAD datasets, specifically designed for consistent anomalies, further supported these observations. CoDeGraph outperformed other methods in both classification and segmentation tasks, with improvements exceeding 10% in F1 and AP metrics. These improvements are due to CoDeGraph's ability to locate and remove consistent anomalies from $\mathcal{B}$ when other approaches fail.

**Inconsistent-Anomaly Datasets.** On the MVTec-IA and Visa datasets, CoDeGraph outperformed the text-based zero-shot methods and matched the performance of MuSc, as shown in Table 2. The performance of CoDeGraph and MuSc was nearly identical, as the absence of consistent anomalies prevented the formation of dense communities in the anomaly similarity graph $\mathcal{G}$. This led to minimal patch exclusions, which had a negligible impact on overall performance.

**Analysis of $\mathcal{P}_{\mathbf{ex}}$.** As illustrated in Figure 6a, $\mathcal{P}_{\mathrm{ex}}$ successfully captured areas of consistent anomalies. On datasets without consistent anomalies, $\mathcal{P}_{\mathrm{ex}}$ hardly took away any patches from the base set $\mathcal{B}$, as shown in Figure 6b. This behavior is validated quantitatively by Table 3. On MVTec-CA, CoDeGraph excluded 6.9% of patches from $\mathcal{B}$ while capturing 73.9% of consistent-anomaly patches. For the ConsistAD dataset, it removed 4.8% of patches and identified 55.2% of total consistent anomalies. In contrast, exclusion rates were minimal for inconsistent-anomaly datasets, reported at 0.3% for MVTec-IA and 0.05% for Visa, preserving the integrity of $\mathcal{B}$, thus maintaining robust performance across diverse datasets.

**Qualitative Results.** Visualizations of anomaly score masks, as shown in Fig. 6, demonstrated CoDeGraph's superior segmentation of consistent anomalies on consistent-anomaly datasets. CoDeGraph's masks fully captured regions with consistent anomalies, such as flipped metal nuts and missing-component cables, while MuSc and other zero-shot methods mostly highlighted edges, failing to cover the full extent. For inconsistent anomalies, CoDeGraph maintained MuSc's precision in segmenting small defects, producing masks with fewer false positives compared to other zero-shot methods.

## 4.3 Ablation Study

**Impact of outlier community detection and patch filtering.** We tested IQR vs. top-3 community selection on $\mathcal{S}_c$ with/without patch filtering. The results are presented in Table 4. On MVTec-CA, performance remained stable, with pixel-wise F1 decreasing from 73.8% to 73.5%. On MVTec-IA, where consistent anomalies are rare, selecting the top-3 communities without patch filtering reduced pixel-wise F1 by 2.9% and image-wise AUROC by 2.0%, as it automatically removed a significant number of patches from $\mathcal{B}$ without selection. Algorithm 2 mitigated this negative impact by removing only high-dependency patches. We also conducted an ablation study on the Tukey's fences parameter $k_{\mathrm{IQR}}$, detailed in Appendix B.3. Notably, for standard values $k_{\mathrm{IQR}} = 1.5$ and $k_{\mathrm{IQR}} = 3$ instead of $k_{\mathrm{IQR}} = 4.5$, our method's performance remained nearly unchanged. Together, outlier detection for outlier communities and targeted patch filtering serve as safeguards, ensuring CoDeGraph's robustness.

---

[2]In the full-shot setting, PatchCore-1 recorded a 98.6% AUROC-cls.

Table 2: Quantitative comparisons on the **Inconsistent-Anomaly Dataset**. We compared CoDeGraph with state-of-the-art zero-shot methods. Bold indicates the best performance. All metrics are in %.

| Dataset | Method | AUROC-cls | F1-cls | AP-cls | AUROC-seg | F1-seg | AP-seg | PRO-seg |
|---------|--------|-----------|--------|--------|-----------|--------|--------|---------|
| MVTec-IA | AnomalyCLIP | 94.1 | 94.0 | 97.5 | 93.4 | 41.6 | 37.1 | 83.1 |
| | WinCLIP | 92.8 | 93.4 | 96.8 | 88.2 | 33.8 | - | 69.1 |
| | APRIL-GAN | 87.9 | 90.9 | 93.5 | 91.7 | 47.5 | 45.4 | 44.3 |
| | ACR | 88.2 | 92.8 | 94.3 | 93.8 | 44.7 | 40.1 | 74.2 |
| | MuSc 30% | 98.0 | **97.6** | 99.0 | **98.2** | 63.7 | 63.8 | 94.0 |
| | MuSc 10% | **98.3** | 97.3 | **99.1** | **98.2** | 64.9 | 65.7 | 94.3 |
| | CoDeGraph (Ours) | **98.3** | 97.3 | **99.1** | **98.2** | **65.1** | **65.8** | **94.4** |
| Visa | WinCLIP | 78.1 | 79.0 | 81.2 | 79.6 | 14.8 | - | 56.8 |
| | APRIL-GAN | 78.0 | 78.7 | 81.4 | 94.2 | 32.3 | 25.7 | 86.8 |
| | MuSc 10% | **91.6** | **89.1** | **92.2** | **98.7** | **48.3** | **45.4** | **91.4** |
| | CoDeGraph (Ours) | **91.6** | 89.0 | **92.2** | **98.7** | **48.3** | **45.4** | **91.4** |

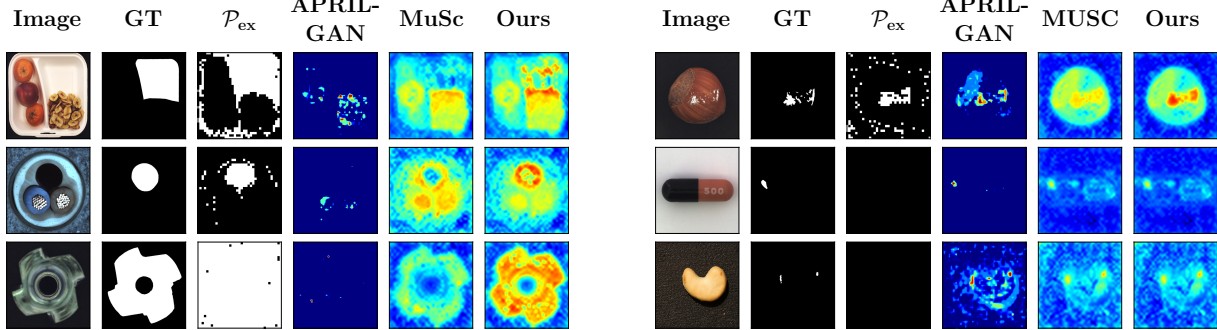

(a) Objects with consistent anomalies    (b) Objects with inconsistent anomalies

Figure 6: Visualization of anomaly segmentation results and excluded patches $\mathcal{P}_{\text{ex}}$.

**Impact of weighted endurance ratio.** As shown in Figure 7, the weighted endurance ratio $\alpha$ shows a strong correlation between consistent-anomaly capture rate and AS performance. For $\alpha > 0.5$, capture rate drops cause a decline in pixel-wise F1, while within the range $\alpha \in [0.0, 0.5]$, both metrics remained stable across the datasets. On ConsistAD, which exhibits high normal-pattern variability, both capture rate and AS improved as $\alpha$ increased, demonstrating the power of $\alpha$ in enhancing the distinctiveness of consistent-anomaly communities in such datasets. In contrast, on inconsistent-anomaly datasets like MVTec-IA, both metrics remained constant across all $\alpha$ values.

**Necessity of Coverage-based Selection.** Coverage-based selection ensures the graph $\mathcal{G} = (V, E)$ provides a comparative baseline for the IQR detection of outlier communities. In our experiments, most datasets required no more than a total of $\binom{N}{2}$ links added to $S_l$ to achieve sufficient coverage. However, the approach of fixing the total number of links in $S_l$ becomes problematic when facing datasets of which patches in consistent-anomaly images are majority consistent anomalies, such as flipped `metal_nut`. To understand this problem, consider a small example: in a test dataset with $q$ duplicated anomalous images, each with $M$ patches, zero-distance links dominate if $\mathcal{S}_l$ contains fewer than $M \cdot \binom{q}{2}$ links. This creates a graph with only connections between the duplication images, offering no baseline for IQR-based detection. We tested CoDeGraph with fixed numbers of links added to $S_l$ on `metal_nut`. For $|S_l| = \binom{N}{2}$, the coverage was 54.7%, while for $|S_l| = \binom{N}{2}/2$, the coverage dropped to 33.9%. In both cases, CoDeGraph failed to identify consistent-anomaly communities as outliers, demonstrating the necessity of coverage-based selection.

**Impact of reference index $\omega$.** An effective $\omega$ selection should enable normal patches to detect similar matches, while consistent anomalies fail to do so. Extreme values of $\omega$, set at $10\%N$ and $90\%N$, degraded performance significantly, as detailed in Table 6. At $\omega = 10\%N$, consistent-anomaly patches, such as flipped `metal_nut` (account for 20% of the test images), easily found similar matches, while at $\omega = 90\%N$, normal

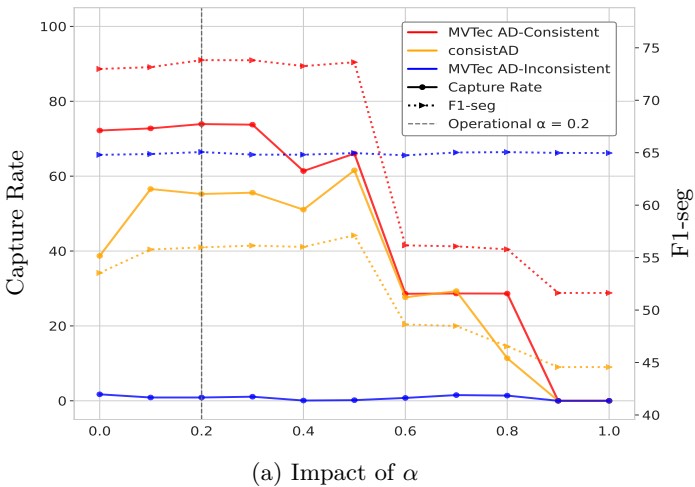

(a) Impact of $\alpha$

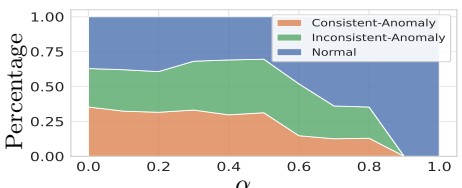

(b) Distribution of $\mathcal{P}_{ex}$ according to $\alpha$

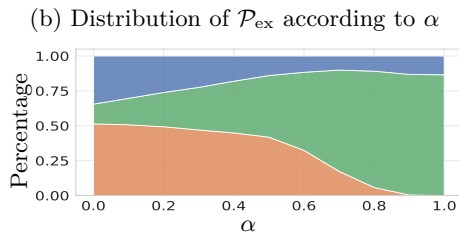

(c) Distribution of $S_l$ according to $\alpha$

Figure 7: Ablation study on the weight endurance ratio parameter $\alpha$. (Left) Impact of $\alpha$ on capture rate and F1-seg performance. (Right) Impact of $\alpha$ on $\mathcal{P}_{ex}$ and suspicious link $S_l$ on MVTec-CA.

**Table 3:** Analysis of $\mathcal{P}_{ex}$, showing exclusion rate, consistent-anomaly capture rate, and patch distribution in (**N**ormal / **C**onsistent / **I**nconsistent). All metrics in %.

| Dataset | Excluded Rate | Capt. Rate | Distribution (N/C/I) |
|---|---|---|---|
| MVTec-CA | 6.9 | 73.9 | 39.3/31.5/29.2 |
| ConsistAD | 4.8 | 55.2 | 39.8/39.6/20.6 |
| MVTec-IA | 0.3 | 0.9 | 92.7/1.7/5.6 |
| Visa | 0.05 | 0.0 | 100.0/0.0/0.0 |

**Table 4:** Ablation study on the impact of outlier community detection methods (IQR vs Top-3) with/without patch filtering, reporting consistent anomalies capture rate, percentage of normal patches removed from $\mathcal{B}$, AUROC-cls, and F1-seg. All metrics are in %.

| Dataset | Method | Filter | Capt. Rate | Removed Normal | AUROC-cls | F1-seg |
|---|---|---|---|---|---|---|
| MVTec-CA | IQR | ✔ | 73.9 | 2.7 | 98.5 | 73.8 |
| | IQR | ✘ | 90.9 | 8.9 | 98.4 | 73.8 |
| | Top-3 | ✔ | 73.9 | 2.8 | 98.5 | 73.8 |
| | Top-3 | ✘ | 90.9 | 12.5 | 98.4 | 73.5 |
| MVTec-IA | IQR | ✔ | 0.9 | 0.2 | 98.3 | 65.0 |
| | IQR | ✘ | 4.9 | 3.2 | 98.2 | 64.9 |
| | Top-3 | ✔ | 2.0 | 3.4 | 96.8 | 63.2 |
| | Top-3 | ✘ | 21.0 | 18.7 | 96.3 | 62.1 |

**Table 5:** Ablation study on inference time on MVTec-AD. $s$ is the number of chunks, and $\eta$ is the percentage of nearest images joining similarity search. All experiments use one RTX 4070Ti Super. All metrics are in %.

| Method | $s$ | $\eta$ | Time (ms) | AUROC-cls | F1-seg |
|---|---|---|---|---|---|
| MuSc | 1 | 1 | 280.46 | 97.5 | 62.3 |
| CoDeGraph | 1 | 1 | 281.34 | 98.3 | 66.8 |
| CoDeGraph | 1 | 0.6 | 211.76 | 98.4 | 66.0 |
| CoDeGraph | 2 | 1 | 192.17 | 98.1 | 66.1 |
| CoDeGraph | 2 | 0.6 | 158.32 | 98.1 | 65.7 |

**Table 6:** Ablation study on reference index $\omega$ on MVTec-CA. All metrics are in %.

| $\omega$ | AUROC-cls | F1-seg |
|---|---|---|
| 0.1N | 94.1 | 56.1 |
| 0.3N | 98.5 | 73.8 |
| 0.5N | 98.5 | 73.3 |
| 0.7N | 98.5 | 73.3 |
| 0.9N | 94.11 | 55.9 |

patches struggled to locate counterparts. Both violated the principles of neighbor-burnout, leading to poor AC/AS. Apart from these extreme values, $\omega$ values between 30% and 70% of $N$ ensure reliable performance. However, for objects with inherent multi-modal variations (e.g., `juice_boxes` in MVTec LOCO have three types of juices), the selection of $\omega$ requires more careful consideration, as legitimate normal variations may be mistakenly identified as consistent anomalies. We provide detailed guidelines for handling multi-modal industrial objects in the Appendix B.2.

**Inference Time and Memory Cost.** CoDeGraph adds negligible processing time compared to MuSc, as shown in Table 5. However, like MuSc, its inference time and GPU memory costs increase with larger test sets due to similarity search demands. CoDeGraph even requires additional VRAM due to operations involving mutual similarity indices. To address this, we adopt MuSc's subset division strategy (Li et al.,

2024), processing test images in $s$ equal-sized chunks. Additionally, we propose CLS token-based screening to reduce similarity search space, where patches in an image $A$ are compared only to patches in $A$'s nearest neighbor images, determined by CLS token similarity and controlled by the nearest neighbor fraction $\eta$. Table 5 shows that combining chunk division $s$ and CLS token screening $\eta$ reduces inference time while maintaining performance. Further discussions are provided in Appendix B.4 and Appendix B.5.

## 5 Related Works

**Zero-shot anomaly detection.** Zero-shot AC and AS leverage foundation models (Dosovitskiy et al., 2021; Radford et al., 2021; Caron et al., 2021; Kirillov et al., 2023) to handle unseen anomalies without training data. WinCLIP (Jeong et al., 2023) introduces text prompts for anomaly segmentation but requires hand-crafted prompts. APRIL-GAN (Chen et al., 2023) proposes learnable projection for image-text alignment, while AnomalyCLIP (Zhou et al., 2023) improves robustness by using object-agnostic learnable prompts. Both APRIL-GAN and AnomalyCLIP depend on auxiliary datasets for training. Similarly, ACR (Li et al., 2023) requires tuning on target-domain-relevant auxiliary data. Drawing inspiration from the rich information in unlabeled datasets, which are successfully used in medical image analysis (Cai et al., 2023; Yoon et al., 2021), MuSc (Li et al., 2024) utilizes unlabeled test images for zero-shot AC/AS. MuSc exploits the characteristic that, in industrial images, normal patches exhibit significant similarity across test sets, whereas anomalous patches do not. However, existing zero-shot methods struggle with consistent anomalies, and CoDeGraph addresses this by using graphs to identify and remove them from anomaly score calculation.

**Graph in machine learning** Graphs have become a cornerstone in machine learning, with Graph Neural Networks (GNNs) (Kipf & Welling, 2016; Hamilton et al., 2017; Veličković et al., 2018) being the most notable example. GNNs have revolutionized the way we interact with complex structured data (Wu et al., 2021), from social networks Ying et al. (2018) to complex physical systems (Sanchez-Gonzalez et al., 2020; Tsubaki & Mizoguchi, 2020). The interaction of graph theory and deep learning extends beyond GNNs. In the literature of information retrieval, pre-trained deep learning models like BERT (Devlin et al., 2019) are used to extract representations for constructing graph databases (Yang et al., 2019; Melnyk et al., 2021; Liu et al., 2020; Wang et al., 2025). In Retrieval-Augmented Generation systems, more researchers use LLMs for knowledge graph extraction (Ranade & Joshi, 2023; Ban et al., 2023; Trajanoska et al., 2023; Yao et al., 2025). GraphRAG (Edge et al., 2024) leverages LLMs to extract entities and relationships from textual data, thereby generating graphs with entities represented as nodes and relationships as edges. Community detection algorithms then divide the graphs into closely related entities to enhance query processing performance. Similarly, CoDeGraph uses patch tokens from ViT to construct graphs that encode relationships between images sharing consistent anomaly patterns. Through community detection on these graphs, CoDeGraph successfully locates images with consistent anomalies.

## 6 Conclusion

This paper introduced CoDeGraph, a zero-shot framework that successfully solves the problem of consistent anomalies by analyzing their similarity growth dynamics. Unlike the predictable growth of normal patches, consistent anomalies exhibit a unique neighbor-burnout phenomenon. CoDeGraph operationalizes this insight by constructing an image-level similarity graph using a principled endurance ratio metric, which makes these hidden recurring patterns explicit. Through community detection, the framework robustly identifies and filters these deceptive anomalies, correcting a key failure point of prior methods. The result is a powerful zero-shot system that advances the state-of-the-art on consistent-anomaly benchmarks while maintaining competitive performance on traditional datasets, demonstrating its value as a general-purpose solution.

**Limitations.** CoDeGraph's applicability is limited when consistent anomaly images outnumber normal images, as may occur in multimodal test datasets. In such cases, the fundamental assumption that normal patterns are more frequent than anomalous ones is violated, and the method may fail to distinguish between legitimate variations and anomalies. Further discussion is presented in Appendix B.2.

**Broader Impact Statement**

This work focuses on industrial anomaly detection for quality control and manufacturing. The proposed method could improve product quality and safety by better detecting defects. However, automated inspection systems should maintain human oversight in critical applications.

**Acknowledgments**

This research was supported by Basic Science Research Program through the National Research Foundation of Korea (NRF) funded by the Ministry of Education (No. RS-2023-00244515).

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

# A  Mathematical Foundations

In this section, we provide theoretical analysis on the similarity growth rate and the coverage-based selection algorithm. Furthermore, we use the Poisson process to explain why consistent-anomaly communities achieve higher densities.

## A.1  A Model for Mutual Similarity Growth Dynamics

This section provides a theoretical model to explain the neighbor-burnout phenomenon—the key empirical observation motivating our work. Using tools from Extreme Value Theory (EVT) Beirlant et al. (2006); Haan & Ferreira (2006); Nair et al. (2022), we demonstrate why the similarity growth for normal patches follows a stable power-law decay, while consistent anomalies deviate sharply from this baseline. This model is intended to provide intuition for our empirical findings and relies on simplifying assumptions, such as treating patch features as independent and identically distributed (i.i.d.), to ensure tractability.

Our analysis begins by assuming a power-law tail for patch-to-patch similarity (validated empirically in Fig. 8). This assumption, under EVT, implies a Fréchet distribution for patch-to-image similarity. This allows us to model normalized distances with a Beta distribution, leading to the main result: the predictable power-law decay of the similarity growth rate for normal patches (Theorem A.1). This formalizes the neighbor-burnout of consistent anomalies as a statistically significant deviation from the norm.

### A.1.1  Patch-to-Patch Similarity and Power-Law Hypothesis

Let $\mathcal{D}$ represent the set of normal patches for a specific object class, and let $x$ be a fixed reference patch. We define $X$ as a random variable representing the distance from $x$ to a normal patch in $\mathcal{D}$, and the patch-to-patch similarity as its reciprocal:

$$S_{\mathrm{p2p}} = \frac{1}{X}.$$

Our core assumption is that $S_{\mathrm{p2p}}$ has a heavy-tailed distribution. Concretely,

*Assumption* A.1 (Asymptotic Power Law Tail). The similarity $S_{\mathrm{p2p}}$ is regularly varying of index $\alpha$. This is formally stated as

$$\mathrm{P}(S_{\mathrm{p2p}} > s) = s^{-\alpha}L(s), \quad \text{as } s \to \infty,$$

where $\alpha > 0$ is the tail index, and $L(s)$ is a slowly varying function.

This assumption is empirically supported by Hill plots (Fig. 8). The Hill plots show stable plateaus for $S_{\mathrm{p2p}}$ across different patches, which strongly suggest that the tail of $S_{\mathrm{p2p}}$ approximately follows a power law.

### A.1.2  Patch-to-Image Similarity via Extreme Value Theory

For an image $I$ with $N$ patches $\{p_1, \ldots, p_N\}$, the distance from $x$ to $I$ is defined as follows:

$$Y = \min_{p \in I} X_p,$$

where $X_p$ is the distance from $x$ to a patch $p$. The patch-to-image similarity is

$$S_{\mathrm{p2i}} = \frac{1}{Y} = \max\left(\frac{1}{X_{p_1}}, \ldots, \frac{1}{X_{p_N}}\right).$$

Thus, $S_{\mathrm{p2i}}$ is the maximum of $N$ patch-to-patch similarities with i.i.d. distribution $F_{S_{\mathrm{p2p}}}$. Despite spatial correlations among patches in an image, the large number of patches (e.g., $N = 1369$) justifies an i.i.d. approximation for distant patch pairs, enabling the application of the Fisher-Tippett-Gnedenko theorem Fisher & Tippett (1928).

This theorem states that for i.i.d. random variables $X_1, \ldots, X_n$ with distribution $F$, the normalized maximum

$$\frac{\max(X_1, \ldots, X_n) - b_n}{a_n} \xrightarrow{d} G,$$

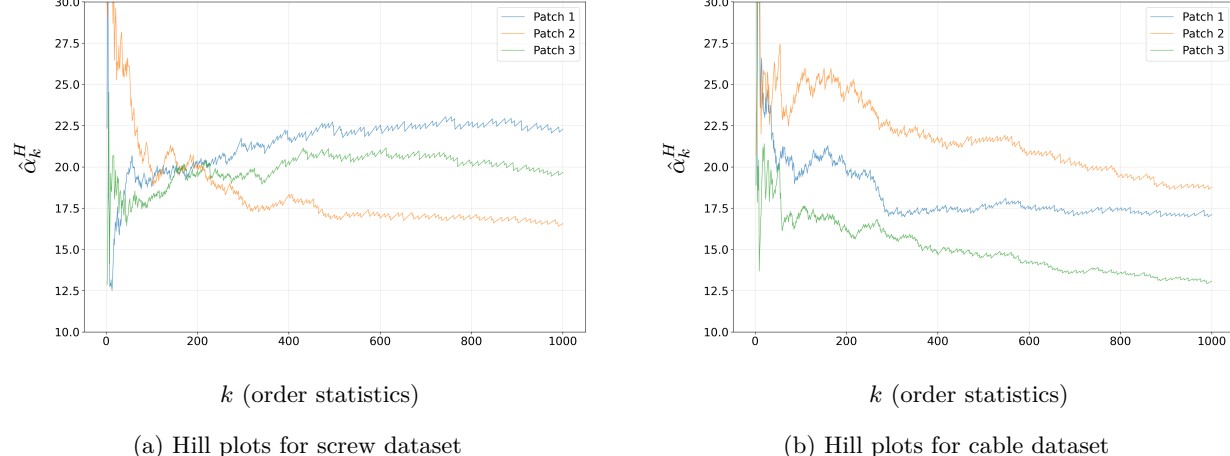

(a) Hill plots for screw dataset

(b) Hill plots for cable dataset

Figure 8: Hill plots showing the tail index estimation for patch-to-patch similarity distributions in industrial datasets. We randomly selected 3 patches from each dataset and computed distances from these patches to all patches in different images. Each plot shows 3 line plots corresponding to the 3 selected patches. The stable plateau in both plots indicates the presence of power-law tails, supporting our Assumption A.1 that the tail of the distribution of $S_{p2p}$ approximately follows a power law.

if and only if $G$ is max-stable. If $1 - F(x) = x^{-\alpha}L(x)$, then $G$ is a Fréchet distribution $\Phi_\alpha$ with tail index $\alpha$, i.e., $P(G > x) \sim x^{-\alpha}$. Given Assumption A.1, $S_{\text{p2i}}$ follows a Fréchet distribution with tail index $\alpha$, implying:

$$P(S_{\text{p2i}} > s) \sim s^{-\alpha}, \quad \text{as } s \to \infty.$$

### A.1.3 Normalized Distance and Beta Distribution

The power-law tail of $S_{\text{p2i}}$ implies a specific distribution for the distance $Y = 1/S_{\text{p2i}}$. For $y \to 0$:

$$P(Y \le y) = P\left(S_{\text{p2i}} \ge \frac{1}{y}\right) \approx Cy^\alpha.$$

To model this behavior, we define a normalized distance index.

**Definition A.1** (Normalized Distance Index). Let $s_0$ be a scale threshold within the power-law region of $Y$. The normalized index $Z$ is:

$$Z = \left(\frac{Y}{s_0} \mid Y \le s_0\right).$$

The CDF of $Z$ is

$$F_Z(z) = P(Z \le z) = \frac{P(Y \le zs_0)}{P(Y \le s_0)} = \frac{C(zs_0)^\alpha}{Cs_0^\alpha} = z^\alpha, \quad z \in [0, 1].$$

This corresponds to a Beta$(\alpha, 1)$ distribution with density $f_Z(z) = \alpha z^{\alpha-1}$.

*Assumption* A.2 (Beta Distribution Model). The normalized patch-to-image distance $Z$ follows a Beta$(\alpha, 1)$ distribution.

Empirical validation confirms this model. We select the endurance ratio $Y_{(i)}/Y_{(\omega)}$ for $i < \omega$, where $\omega$ is a low-order statistic (e.g., $\omega \approx 0.3N$). This choice ensures $Y_{(\omega)}$ lies within the power-law tail region of the distance distribution, where the Beta$(\alpha, 1)$ model is most applicable. Q-Q plots (Fig. 9) demonstrate that $Y_{(i)}/Y_{(\omega)}$ aligns closely with a Beta$(\alpha, \beta)$ distribution, with $\beta \approx 1$ and $\alpha \gg \beta$, supporting our theoretical works.

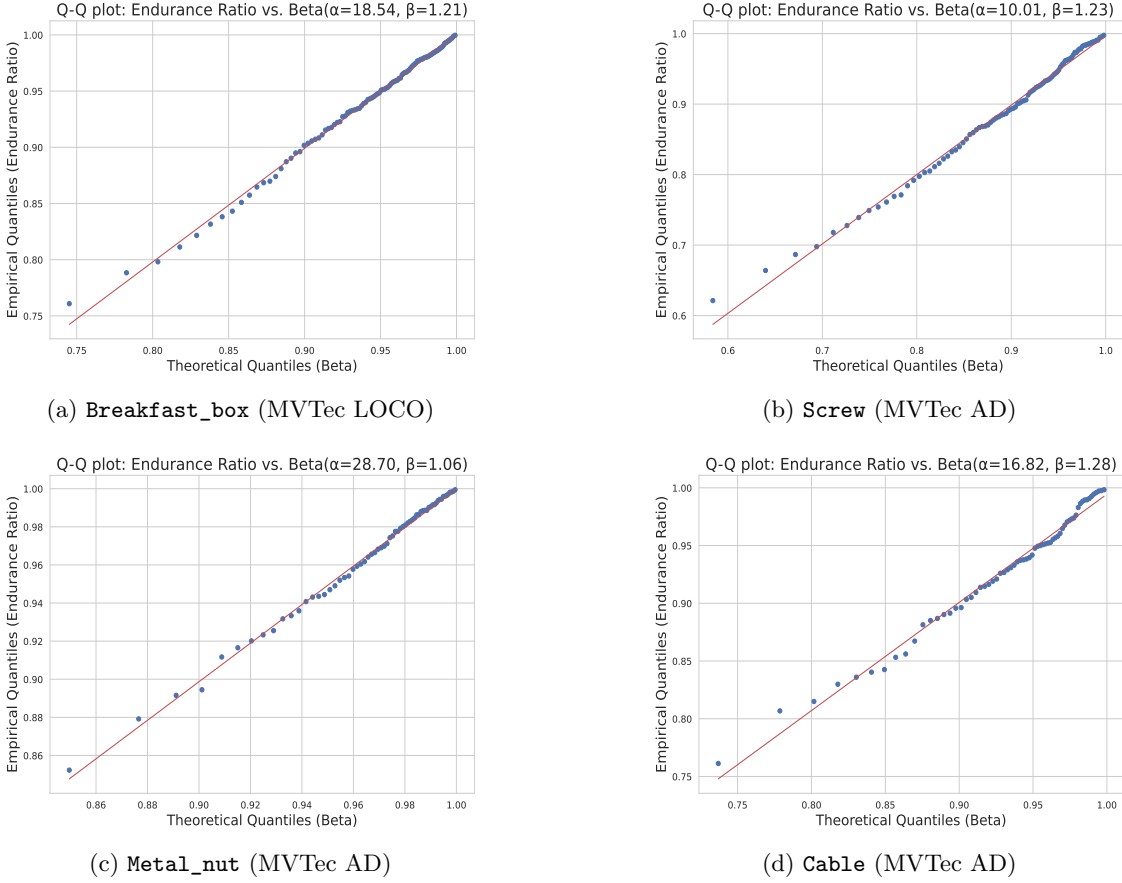

(a) `Breakfast_box` (MVTec LOCO)  (b) `Screw` (MVTec AD)

(c) `Metal_nut` (MVTec AD)  (d) `Cable` (MVTec AD)

Figure 9: Q-Q plots validating the Beta distribution assumption for similarity indices across four normal industrial objects, including objects with inconsistent normal patterns (`screw`, `breakfast_box`). For each object, we fixed one randomly selected patch and computed similarity indices from this patch to all patches in different images. Each plot compares the empirical quantiles of $X_{(i)}/X_{(\omega)}$ with $\omega = 0.3N$ against theoretical quantiles from its fitted Beta distribution. The close alignment along the diagonal demonstrates that our modeling assumption is reasonable.

### A.1.4 Similarity Growth Rate and Log-Spacing

Let $Y_{(1)} \leq \cdots \leq Y_{(\omega)}$ be the order statistics of $\omega$ i.i.d. samples of $Y \mid Y < s_0$, with $Y_{(i)} = s_0 Z_{(i)}$ and $Z_{(i)}$ the order statistics of $Z \sim \text{Beta}(\alpha, 1)$. The similarity growth rate is defined as

$$\tau^{(i)}(x) = \ln \frac{Y_{(i+1)}}{Y_{(i)}}.$$

We derive its statistical properties using two lemmas.

**Lemma A.1.** *If* $Z \sim \text{Beta}(\alpha, 1)$, *then* $-\ln Z \sim \text{Exp}(\alpha)$.

*Proof.* For $Z \sim \text{Beta}(\alpha, 1)$, the density is $f_Z(z) = \alpha z^{\alpha-1}$, $z \in (0, 1)$. Let $W = -\ln Z$. The density of $W$ is

$$f_W(w) = f_Z(e^{-w}) \cdot \left| \frac{dz}{dw} \right| = \alpha(e^{-w})^{\alpha-1} \cdot e^{-w} = \alpha e^{-\alpha w}, \quad w > 0,$$

which is the density of an $\text{Exp}(\alpha)$ distribution. □

**Lemma A.2.** *For i.i.d.* $W_1, \ldots, W_n \sim \text{Exp}(\lambda)$, *the spacings* $W_{(k+1)} - W_{(k)}$ *are independent and follow* $\text{Exp}(\lambda(n-k))$ .

*Proof.* This is a well-established result in order statistics theory (David & Nagaraja, 2004). □

**Theorem A.1** (Log-Spacing of Order Statistics). *The log-spacings* $\ln Y_{(i+1)} - \ln Y_{(i)}$ *are independent and follow* $\text{Exp}(\alpha i)$ *for* $i = 1, \ldots, \omega - 1$.

*Proof.* Let $Z_1, \ldots, Z_\omega \sim \text{Beta}(\alpha, 1)$, and define $W_j = -\ln Z_j$. By Lemma A.1, $W_j \sim \text{Exp}(\alpha)$. Since $g(z) = -\ln z$ is decreasing, $W_{(k)} = -\ln Z_{(n-k+1)}$. For $Y_{(i)} = s_0 Z_{(i)}$, we have

$$\ln Y_{(i+1)} - \ln Y_{(i)} = \ln Z_{(i+1)} - \ln Z_{(i)} = W_{(\omega-i+1)} - W_{(\omega-i)}.$$

Setting $k = \omega - i$, this spacing is $W_{(k+1)} - W_{(k)} \sim \text{Exp}(\alpha(\omega - k)) = \text{Exp}(\alpha i)$ by Lemma A.2. □

This theorem directly leads to the following corollaries about the moments of the growth rate.

**Corollary A.1** (Growth Rate Properties). *The moments of* $\tau^{(i)}(x) = \ln Y_{(i+1)} - \ln Y_{(i)}$ *exhibit power-law decay:*

$$\mathbb{E}[\tau^{(i)}(x)] = \frac{1}{\alpha i}, \quad \text{Var}[\tau^{(i)}(x)] = \frac{1}{(\alpha i)^2}.$$

**Corollary A.2** (Cumulative Growth Properties). *For cumulative growth* $\Delta^{(i,j)}(x) = \ln Y_{(j)} - \ln Y_{(i)}$, $j > i$:

$$\mathbb{E}[\Delta^{(i,j)}(x)] = \frac{1}{\alpha} \sum_{k=i}^{j-1} \frac{1}{k}, \quad \text{Var}[\Delta^{(i,j)}(x)] = \frac{1}{\alpha^2} \sum_{k=i}^{j-1} \frac{1}{k^2}.$$

The variance is bounded by $\frac{\pi^2}{6\alpha^2}$, and $\text{Var}[\Delta^{(i,j)}(x)] \to 0$ as $i, j \to \infty$ with $j - i$ fixed, justifying the stability of the endurance ratio $\zeta$ for anomaly detection.

*Remark* A.1 (Generality of Power-Law Framework). Our theoretical framework applies to any distribution whose tail approximately follows a power law. This generality explains why in empirical log-log plots (such as Fig. 3 and Fig. 10), inconsistent-anomaly patches also exhibit power-law decay in their growth rate, and even consistent-anomaly patches demonstrate similar behavior in early steps before neighbor-burnout occurs. The key distinction lies in the disruption of this power-law pattern for consistent anomalies after exhausting their limited pool of similar matches, which our endurance ratio metric effectively captures.

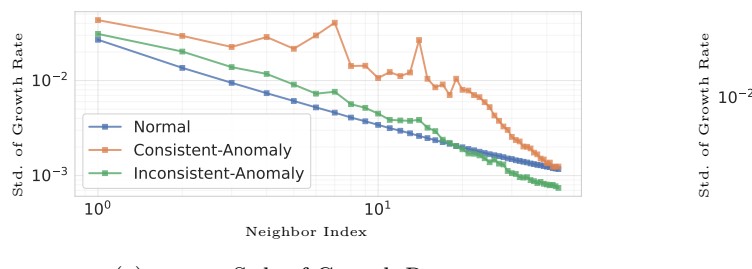

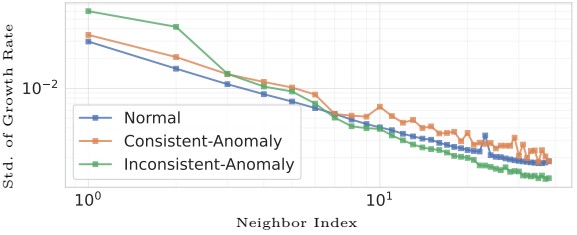

(a) `Cable`: Std. of Growth Rate

(b) `Capsule`: Std. of Growth Rate

Figure 10: Log-log plots of standard deviation of growth rate $\tau^{(i)}(x)$ on the test images of `Cable` and `Capsule`.

## A.2 Stopping Time of Coverage-Based Selection

A primary concern in coverage-based selection is whether the incremental link addition process might excessively connect non-consistent-anomaly (non-CA) nodes, thereby diluting the signal from consistent-anomaly nodes. To analyze this risk, we examine the expected number of links required to achieve the target coverage $\tau$, where every non-CA node attains a degree of at least one ($d(v) \geq 1$).

We model our coverage-based selection using the **Generalized Coupon Collector Problem** (Sellke, 1995; Johnson & Sellke, 2010), which can be visualized as observing all balls in an urn. Consider an urn containing

$n$ distinct white balls (numbered $1, 2, \ldots, n$). In each turn: (1) a random sample of size $K$ balls is drawn without replacement, (2) any white balls in the sample are painted red, and (3) all balls are returned to the urn. The process repeats until all balls are red. The generalized coupon collector problem seeks to determine the expected waiting time (number of turns) until completion.

**Mapping to Coverage-Based Selection:** Non-CA nodes correspond to white balls. A non-CA node achieving $d(v) \geq 1$ corresponds to its ball being painted red. Each link addition step constitutes one turn. Adding a link from a CA node to a non-CA node corresponds to drawing a sample of size $K = 1$. Adding a link between two non-CA nodes corresponds to drawing a sample of size $K = 2$.

**Analysis and Bounds:** The classical coupon collector result (where $\mathbb{P}(K = 1) = 1$) provides an upper bound for $\mathbb{E}[\tau]$ in our mixed $K$ setting, since a $K = 2$ step covers two nodes simultaneously, accelerating coverage compared to always using $K = 1$. For $n$ balls where ball $j$ is collected with probability $p_j$ per turn, the expected number of turns satisfies (Ross, 2010):

$$\mathbb{E}[T] = \int_0^\infty \left[ 1 - \prod_{j=1}^n (1 - e^{-p_j t}) \right] dt. \tag{5}$$

In the idealized case where balls are drawn with equal probability, this yields the classical result $\mathbb{E}[T] = n \cdot H_n$, where $H_n$ is the $n$-th harmonic number. However, in practice, although non-CA nodes receive links in a relatively balanced manner, outlier images may exist where the probability of adding a link to them is extremely small. When $p_{\min} = \min_i p_i$ approaches zero, by Fatou's lemma we obtain $\mathbb{E}[T] \to \infty$. The partial coverage threshold $\tau$ prevents this infinite link addition to the graph $\mathcal{G}$ by allowing the ignoring of certain nodes that are difficult to reach.

Suppose there are $m$ non-CA nodes remaining, and let $m \cdot p_{\min}^{(m)} = 1/C$ where $p_{\min}^{(m)}$ is the minimum probability among the remaining nodes. By the change of variables,

$$\mathbb{E}[T] \leq \int_0^\infty \left[ 1 - \left( 1 - e^{-p_{\min}^{(m)} t} \right)^m \right] dt = \frac{H_m}{p_{\min}^{(m)}} \sim C \cdot m \ln m. \tag{6}$$

Since hard-to-reach nodes were ignored through the threshold $\tau$, we might assume that $C$ is not extremely large, thus providing a reasonable upper bound. If we further assume that $m$ non-CA nodes are selected uniformly at random in each turn, then under this assumption, the expected number of links added follows the approximation derived in (Johnson & Sellke, 2010). Let $p$ denote the probability that an added link connects two non-CA nodes, and $(1 - p)$ the probability that it connects a CA node to a non-CA node, i.e., $P(K = 1) = 1 - p$ and $P(K = 2) = p$. Then:

$$\mathbb{E}[T] = \frac{H_m}{a_1} + \frac{a_2}{a_1^2} + \mathcal{E}_m,$$

where the coefficients are,

$$a_1 = \sum_{r=0}^{m-1} \frac{\mathbb{P}(K > r)}{m - r} = \frac{1}{m} + \frac{p}{m - 1},$$

$$a_2 = \sum_{r=1}^{m-1} \frac{\mathbb{P}(K > r)}{m - r} \sum_{j=1}^r \frac{1}{m - j + 1} = \frac{p}{m(m - 1)}.$$

The error term is bounded: $|\mathcal{E}_m| \leq C_0 e^{-2m/3}$ for some constant $C_0 > 0$ that does not depend on $m$. For large $m$, this yields the asymptotic behavior:

$$\mathbb{E}[T] \sim \frac{m \ln m}{1 + p} + \frac{p}{(1 + p)^2} + \mathcal{O}(e^{-2m/3}). \tag{7}$$

Both inequalities equation 6 and equation 7 demonstrate that coverage-based selection exhibits at maximum $\mathcal{O}(m \ln m)$ links for images that are not consistent-anomaly images, while they are broadly distributed

among at least $\binom{m}{2}$ edges. This ensures that coverage-based selection remains computationally efficient and maintains the sparse connectivity essential for subsequent community detection.

### A.3 Community Density

In this section, we study the density of communities by modeling CoDeGraph's link addition process as a Poisson process (N(t)) with a rate of $\lambda = 1$. Each incoming link is classified into one of $k + 1$ communities: $C_1, \ldots, C_k$ are the communities naturally formed by the nature of the dataset, and the $(k+1)$-th community represents inter-community connections.

By Theorem 5.2 in Ross (2010), let $N_j(t)$ denote the number of $C_j$ links added by time $t$. Then $\{N_j(t)\}$ are independent Poisson processes with rate $\lambda \cdot p_j = p_j$, where $p_j$ is the probability that a link belongs to community $j$, and $\sum_{j=1}^{k+1} p_j = 1$. At stopping time $T$ determined by the coverage-based selection criterion, the expected number of links added to community $j$ is

$$\mathbb{E}[N_j(T)] = T \cdot p_j.$$

For communities $C_1, \ldots, C_k$ with sizes $m_j = |C_j|$, the expected density of community $C_j$ is given by

$$\mathbb{E}[\rho(C_j)] = \frac{\mathbb{E}[N_j(T)]}{m_j(m_j - 1)} = \frac{T \cdot p_j}{m_j(m_j - 1)}.$$

Due to the neighbor-burnout phenomenon, consistent-anomaly communities accumulate links at a significantly higher rate during the stopping period $T$ of coverage-based selection, whereas other communities accumulate links at a slower pace. This leads to an expected weight density that is considerably higher for consistent-anomaly communities than for non-CA communities. Hence, consistent-anomaly communities become detectable as density outliers in the graph structure.

## B Additional Ablation Study

### B.1 Community Detection Algorithms

We investigate the impact of different community detection algorithms on CoDeGraph.

**Modularity-Based Community Detection.** One of the most popular methods in community detection is called modularity (Newman & Girvan, 2004). Modularity is proposed specifically to measure the strength of a community partition for real-world networks by taking into account the degree distribution of nodes. It partitions the graph by maximizing the difference between observed edge weights and expected edge weights under a null model. The modularity objective is defined as follows:

$$Q = \sum_{ij} \left( A_{ij} - \frac{k_i k_j}{2m} \right) \delta(\sigma_i, \sigma_j),$$

where $A_{ij}$ is the adjacency matrix, $k_i$ and $k_j$ are the degrees of nodes $i$ and $j$, $m$ is the total edge weight in the graph, and $\delta(\sigma_i, \sigma_j) = 1$ if nodes $i$ and $j$ belong to the same community, and 0 otherwise. Here, $k_i k_j / 2m$ is the expected number of edges between nodes $v_i$ and $v_j$.

**Limitations of Modularity on Anomaly Similarity Graphs.** While modularity is effective in general-purpose graphs, its degree-based null model introduces a key limitation for the anomaly similarity graph $\mathcal{G}$ constructed in CoDeGraph. In our graph, consistent-anomaly images form high-density communities through many strong connections, but some links may be weaker due to natural intra-class variation (e.g., illumination, pose, or noise). Despite being much stronger than any connection between normal images, these weak intra-community links can fall below the expected weight $k_i k_j / 2m$ implied by node degrees. As a result, modularity treats them as unexpectedly weak and penalizes their presence in the same community. This leads to the **fragmentation of consistent-anomaly clusters** into multiple high-density communities.

This fragmentation undermines our IQR-based community scoring, diluting the density of anomaly clusters and hindering the reliable detection of consistent-anomaly patterns. Figure 11 illustrates this effect on the `metal_nut` class, where semantically identical flip anomalies are split into three separate communities, despite their strong relative similarity.

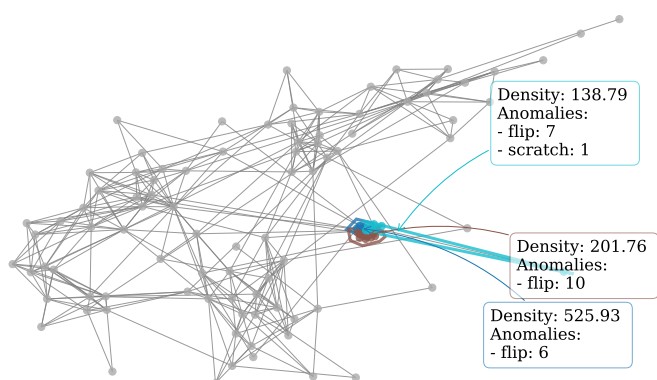

Figure 11: Modularity-based community detection fragments a consistent anomaly type (flipped metal nuts) into several smaller communities due to weak intra-group links falling below degree-based expectations. This fragmentation interferes with reliable detection of consistent-anomaly groups.

**Why CPM is More Appropriate.** CPM avoids these issues by evaluating communities based on absolute resolution $\gamma$, independent of node degree. This makes it well-suited to our anomaly similarity graph, where consistent-anomaly communities may contain a mix of strong and moderately strong links. As long as the overall density exceeds the resolution threshold $\gamma$, the community is preserved intact. As described in the main paper, we set $\gamma$ to the 25th percentile of edge weights. This choice ensures that most connections within consistent-anomaly communities lie above this threshold, hence preventing the fragments. This results in more stable and coherent detection of consistent-anomaly clusters, which is essential for reliable patch filtering and improved segmentation performance in the zero-shot setting.

## B.2 Multi-Modal Industrial Objects

This subsection examines the challenges of applying CoDeGraph to industrial settings with multi-modal normal variations and scenarios where dominant consistent anomalies lead to failure modes.

**Parameter Selection for Multi-modal Normal Variations.** In vast industrial domains, industrial objects may exhibit multi-modal variations that represent different normal states. For example, `juice_bottles` in MVTec LOCO contains three distinct juice types (banana, orange and tomato), each representing a separate normal variant. In such cases, selection of the reference index $\omega$ requires careful consideration to avoid misclassifying normal variations as consistent anomalies. The reference index $\omega$ must satisfy $\omega < N/C$, where $N$ is the total number of test images and $C$ is the number of distinct normal variants. This threshold allows patches within each normal variant to maintain similar neighbors before experiencing the neighbor-burnout phenomenon. When $\omega > N/C$, the similarity graph forms clusters corresponding to different normal variants, as shown in Fig. 12. Thus, the community detection algorithm may then identify normal variations as outlier communities.

**Limitations with Dominant Consistent Anomalies.** CoDeGraph assumes normal patterns are more frequent than anomalies, enabling the identification of consistent anomalies as outlier communities. However, this assumption fails when consistent anomalies outnumber specific normal variants. For instance, if 25% of juice bottles are empty (anomalous) and only 20% contain orange juice, empty bottles may appear more "normal" than orange juice bottles. Consequently, CoDeGraph may misclassify frequent empty bottles

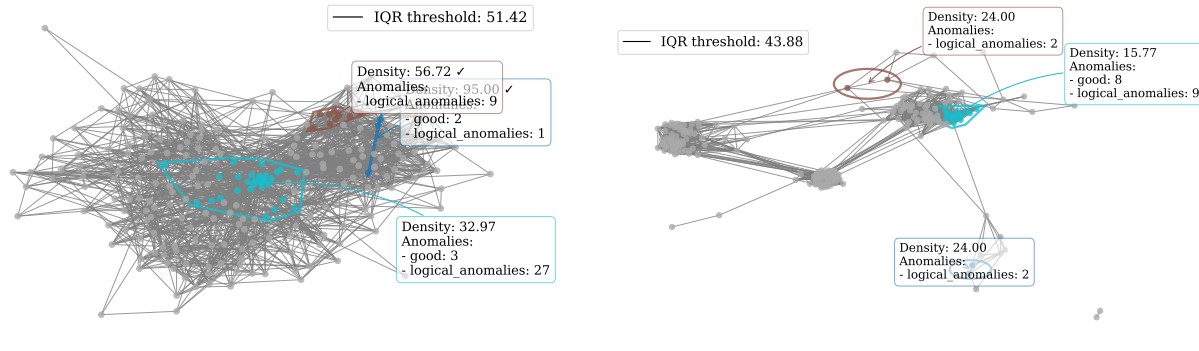

(a) Multi-modal case with $\omega = 10\%$ of $N$        (b) Multi-modal case with $\omega = 50\%$ of $N$

Figure 12: Visualization of multi-modal juice bottle classification. (a) At $\omega = 10\%$ of $N$, connections among anomalous and consistent-anomaly images are clear and detectable by CPM. (b) At $\omega = 50\%$ of $N$, normal variant connections dominate, obscuring anomalous image connections.

as normal and less frequent normal variants (e.g., orange juice) as anomalies, undermining its ability to distinguish between normal and consistent-anomaly patterns.

**Experimental Analysis of Failure Modes.** Due to the absence of multi-modal industrial datasets with dominant consistent anomalies, we designed a controlled experiment using the `juice_bottle` subclass from MVTec LOCO. We selected 8 tomato, 4 orange, and 4 banana juice bottle images as normal variants. To simulate dominant anomalies, we introduced 8 consistent-anomaly images (logical anomalies lacking orange labels, e.g., `logical_anomalies_000`, `003`, `010`, `011`, `072`, `077`, `078`, `079`), outnumbering the orange and banana variants. In the similarity graph (Figure 13), the less frequent orange and banana variants form dense clusters, which can be wrongly identified as anomalous communities.

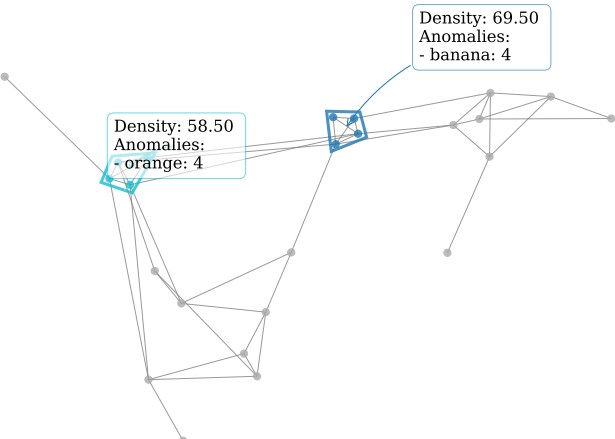

Figure 13: Visualization of anomaly similarity graph when consistent anomalies outnumber specific normal variants.

This experiment underscores the limitation of CoDeGraph when consistent anomalies outnumber specific normal variants, leading to misclassification of legitimate multi-modal normals. In practice, such scenarios are rare, as none of the datasets, such as MVTec AD or Visa, contain more than 20% consistent-anomaly samples. Future work could explore pre-clustering normal variants to isolate them before applying CoDeGraph, enhancing robustness in diverse industrial settings with multi-modal data.

### B.3 Tukey Parameter Selection for IQR Outlier Detection

In CoDeGraph, we use Tukey's fences IQR outlier detection to identify outlier communities. A community is flagged as an outlier if its density exceeds $Q_3 + k_{\text{IQR}} \cdot IQR$, where $Q_3$ is the third quartile and $IQR$ is the interquartile range of the community density distribution. The IQR parameter $k_{\text{IQR}}$ determines how strictly we identify a community as an outlier. In the main paper, we purposefully set $k_{\text{IQR}} = 4.5$, a strict threshold that only extreme far-out outliers could overcome. This extreme choice presents clear empirical evidence that consistent-anomaly images form distinctly dense communities, far exceeding typical statistical outliers, thereby validating the discriminative power of our graph-based approach.

For real-world applications, we recommend more standard IQR-based outlier detection thresholds, such as $k_{\text{IQR}} = 1.5$ or $k_{\text{IQR}} = 3$. These values strike a balance between sensitivity and specificity, effectively capturing a wider range of consistent-anomaly patterns. The rationale for these moderate thresholds is twofold: (1) both $k_{\text{IQR}} = 1.5$ and $k_{\text{IQR}} = 3$ reliably identify outliers while being less restrictive than $k_{\text{IQR}} = 4.5$, and (2) our targeted patch filtering mechanism in Algorithm 2 mitigates the risk of erroneously removing normal patches from the base set $\mathcal{B}$, enhancing robustness. We evaluated the impact of varying $k_{\text{IQR}}$ values on the MVTec AD dataset, with results summarized in Table 7:

Table 7: Ablation study on $k_{\text{IQR}}$ for outlier detection on MVTec AD dataset. All metrics are in %.

| Value | AUROC-cls | F1-cls | AP-cls | AUROC-seg | F1-seg | AP-seg | PRO-seg |
|---|---|---|---|---|---|---|---|
| $k_{\text{IQR}} = 1.5$ | 98.01 | 97.08 | 99.11 | 98.14 | 66.32 | 66.64 | 94.46 |
| $k_{\text{IQR}} = 3.0$ | 98.29 | 97.39 | 99.21 | 98.16 | 66.41 | 66.81 | 94.59 |
| $k_{\text{IQR}} = 4.5$ (default) | 98.32 | 97.39 | 99.24 | 98.20 | 66.82 | 68.06 | 94.59 |

The results demonstrated that both $k_{\text{IQR}} = 3$ and $k_{\text{IQR}} = 1.5$ outperformed all other zero-shot methods reported in the main paper, achieving over 98% AUROC for both classification and segmentation tasks. Importantly, the effectiveness of these smaller $k$ values indicates that practitioners can confidently set $k$ to more sensitive thresholds to capture all potential consistent anomalies while not worrying about performance degradation on general datasets. This flexibility allows CoDeGraph to be more inclusive in identifying outlier communities, ensuring that even moderately cohesive consistent-anomaly patterns are detected without compromising the method's robustness across diverse industrial scenarios.

### B.4 Computational Efficiency and Memory Considerations

The base VRAM memory requirements for MuSc include the ViT model weights, extracted features from all images in the base set $\mathcal{B}$, and the computed distance vectors $D_{\mathcal{B}}^r(x_{i,m}^l)$ for mutual similarity ranking. These components constitute the primary memory allocation for the underlying scoring mechanism. Due to operations that relate to indexing the distance vectors in Algorithm 1 and Algorithm 2, CoDeGraph introduces additional memory overhead through index storage for the distance vectors. These indices are of size $[L, N, M, N-1]$, where $L$ denotes the number of ViT layers, $N$ the number of images, $M$ the number of patches per image, and $N-1$ accounts for excluding self-comparisons, which can be huge for large datasets. However, these indices allow us to efficiently implement the calculation of distance vectors over the new base set $\mathcal{B} \setminus \mathcal{P}_{\text{ex}}$. Rather than recalculating distance vectors over the new base set $\mathcal{B} \setminus \mathcal{P}_{\text{ex}}$, our implementation uses `torch.isin` to identify invalid distances whose indices correspond to excluded patches $\mathcal{P}_{\text{ex}}$. Although this approach allocates more VRAM, these operations help us add minimal processing time, as shown in Table 5.

For GPU-constrained environments, we recommend mapping index tensor operations exclusively to CPU while maintaining feature computations on GPU. Empirical evaluations on a dataset of 200 images showed a maximum VRAM allocation of 7.4 GB (CPU indexing) versus 10.4 GB (full GPU). Processing time evaluations on MVTec AD added minimal computational overhead: average processing time increased from 281 ms to 291 ms per image. For MVTec-CA, which contains several outlier communities requiring multiple iterations in Algorithm 2 (as the algorithm loops over all communities in $\mathcal{S}_c$), the average processing time

increased from 327 ms to 350 ms per image, representing acceptable trade-offs for memory-constrained deployments.

## B.5   CLS Token-based Screening Analysis

Table 8: Extended analysis of CLS token-based screening performance on MVTec AD dataset with different $\eta$.

| $\eta$ | Time (ms) | AUROC-cls (%) | F1-seg (%) |
|--------|-----------|---------------|------------|
| 1.0 | 281.34 | 98.3 | 66.8 |
| 0.8 | 246.33 | 98.2 | 66.5 |
| 0.6 | 211.76 | 98.4 | 66.0 |
| 0.4 | 178.52 | 98.3 | 65.2 |
| 0.2 | 148.62 | 96.7 | 60.5 |

CLS token-based screening is a computational optimization technique that uses the global representation capabilities of Vision Transformer (ViT) CLS tokens to pre-filter patches before applying Mutual Similarity Ranking. We extract CLS tokens from the ViT backbone's final layer, which encode global image-level representations of semantic information and structural patterns. Then, we calculate the cosine similarity between each target picture $I_i$ and other images in the base set $\mathcal{B}$ using CLS tokens. The similarities are ranked to determine the top $\eta \cdot N$ most similar photos, where $\eta \in [0, 1]$ indicates the closest neighbor fraction and $N$ specifies the total number of images in the base set.

This approach is based on the observation that images with similar CLS tokens share similar overall features, like postures. For example, screws with the same rotation have higher CLS token cosine similarity than screws with different postures, or cables with missing components tend to find each other using CLS token similarity. By constraining the Mutual Similarity Ranking process to operate only on patches from these $\eta \cdot N$ similar images, we reduce the computational burden from $O(N^2 M^2)$ to $O(\eta N^2 M^2)$, where $M$ represents the number of patches per image.

The performance of CLS token-based screening across various values of $\eta$ is presented in Table 8. Reduced search space led to decreased processing time, while performance remained stable. At $\eta = 0.4$, the AUROC-cls remained stable, whereas the F1-seg decreased from 66.8% to 65.2%, and the processing time reduced from 281.34 ms to 178.52 ms per image (about 37%). At $\eta = 0.2$, the search space becomes excessively constrained. For example, in the case of `metal_nut`, the detection of neighbor-burnout phenomena requires that $\omega > 0.2N$, meaning the quantity of images contributing to mutual similarity ranking must exceed $0.2N$ for functionality. This resulted in AUROC-cls falling to 96.7% and F1-seg declining to 60.5% for $\eta = 0.2$.

## B.6   Effect of Different Backbones

We investigated the impact of various backbone architectures on CoDeGraph's anomaly detection performance. Our evaluation covered different Vision Transformer architectures such as DINO (Caron et al., 2021), DINOv2 (Oquab et al., 2023), and CLIP (Radford et al., 2021). We used the same four-stage division scheme across all architectures: ViT-Large models were divided into 4 stages with 6 layers each, while ViT-Base models used 3 layers per stage. We extract patch tokens from each stage and utilize the linearly projected class token from the final layer for classification optimization. All other experimental settings remained identical to those described in the main text.

Table 9 presents the comprehensive performance comparison across different backbone architectures. All backbone architectures demonstrated solid anomaly detection performance across both classification and segmentation tasks. Generally, larger models trained with smaller patch sizes and higher resolutions achieved better results, with DINOv2-L-14 showing the best overall performance while CLIP ViT-L-14-336 achieved the highest classification AUROC. These results confirm that CoDeGraph's graph-based approach generalizes effectively across different ViT architectures.

Table 9: Comparison of CoDeGraph performance across different backbone architectures on MVTec AD. Results show AUROC and F1-score for both anomaly classification and anomaly segmentation on the ConsistAD benchmark.

| Pre-training Method | Architecture | Pre-training Dataset | AUROC-cls | F1-cls | AUROC-seg | F1-seg | AP-seg |
|---|---|---|---|---|---|---|---|
| DINO | ViT-B-8 | ImageNet-1k | 97.1 | 97.1 | 98.4 | 65.7 | 67.8 |
| | ViT-B-16 | ImageNet-1k | 96.8 | 96.9 | 98.6 | 66.9 | 69.1 |
| DINOv2 | ViT-B-14 | LVD-142M | 97.6 | 97.5 | 98.7 | 67.1 | 69.3 |
| | ViT-L-14 | LVD-142M | 98.0 | **97.7** | **98.9** | **69.1** | **71.9** |
| CLIP | ViT-B-16 | WIT-400M | 96.2 | 95.8 | 98.1 | 65.9 | 67.0 |
| | ViT-L-14-336 | WIT-400M | **98.3** | 97.4 | 98.2 | 66.8 | 68.1 |

## B.7 Analysis of the Internal Average Parameter

The internal average parameter $K$ in Equation equation 1 controls the interval size for calculating anomaly scores within the Mutual Scoring Mechanism. Finding the optimal value for $K$ is challenging due to its strong dependence on the dataset. In this section, we analyze how $K$ affects performance across different dataset types and demonstrate how CoDeGraph mitigates this parameter sensitivity. This also explains our decision to set $K = 10\%N$ rather than $30\%N$ as in Li et al. (2024).

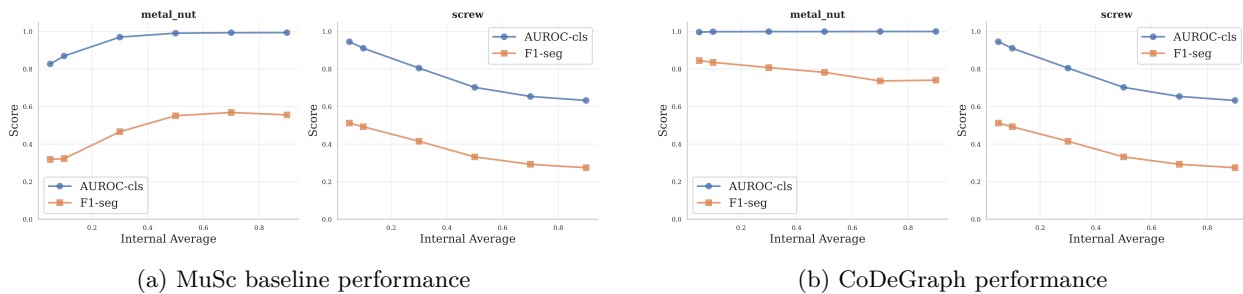

(a) MuSc baseline performance      (b) CoDeGraph performance

Figure 14: Impact of internal average parameter $K$ on anomaly detection performance. (a) MuSc baseline shows opposite behavior for consistent-anomaly datasets (`metal_nut`) versus datasets with inconsistent postures (`screw`), creating parameter selection conflicts. (b) CoDeGraph exhibits a similar relationship between AC/AS performance and $K$ for both dataset types.

**Dataset-Dependent Behavior of Parameter $K$.** The impact of $K$ on performance depends on the underlying dataset characteristics. For consistent-anomaly datasets such as `metal_nut`, increasing $K$ beyond the number of consistent neighbors $H$ improves performance. From the bound derived in Section 3.2, we obtain

$$a_{\mathcal{B}}^r(x_a) \leq \frac{H_0 \epsilon}{K} + \frac{1}{K} \sum_{i=H_0+1}^{K} d^r(x_a, I_{(i)}),$$

where $H_0 = \min(H, K)$. When $K > H$ and $K$ is sufficiently large, the distances beyond $H$ consistent neighbors balance the small distances from deceptive matches, causing $a_{\mathcal{B}}^r(x_a)$ to increase and properly penalize consistent anomalies. This explains why consistent-anomaly patches require larger $K$ values to achieve proper scoring. Figure 14a demonstrates this behavior, where `metal_nut` shows improvement in F1-segmentation scores as $K$ increases. In contrast, for objects with inconsistent postures, such as `screw`, there is an opposite tendency. As $K$ approaches $N$, distances to the K-th nearest neighbor $I_{(K)}$ become less reliable, causing performance drops, as visualized in Fig 14a. This establishes a trade-off in parameter selection, whereby an optimal $K$ for consistent-anomaly datasets yields a suboptimal $K$ for datasets with inconsistent postures.

**CoDeGraph's Robustness.** Since CoDeGraph is able to remove deceptive matches from the base set $\mathcal{B}$, our method performs in a more predictable manner. As shown in Figure 14b, both datasets exhibit a similar relationship between AC/AS performance and the internal average parameter $K$, where the optimal performance is around small $K$ instead of large $K$. Targeted filtering by CoDeGraph makes $H$ near zero for all patches in the refined base set $\mathcal{B} \setminus \mathcal{P}_{\mathrm{ex}}$, eliminating the need for large $K$.

Our analysis demonstrates that CoDeGraph reduces the parameter sensitivity issue, making $K$ selection more robust across diverse datasets.

## B.8 Additional Benchmark Results

We present additional comparisons with few-shot and full-shot methods in this appendix. For few-shot methods, we compared with PatchCore (Roth et al., 2022), WinCLIP (Jeong et al., 2023), APRIL-GAN (Chen et al., 2023), and GraphCore (Xie et al., 2023) in the 4-shot setting. For full-shot methods, we included CFlow (Gudovskiy et al., 2022) and DDAD (Mousakhan et al., 2024). The results are presented in Table 10 and Table 11.

Table 10: Quantitative comparisons on **Consistent-Anomaly Datasets**. We compared CoDeGraph with few-shot and full-shot methods. Bold indicates the best performance. All metrics are in %.

| Dataset | Method | Setting | AUROC-cls | F1-cls | AP-cls | AUROC-seg | F1-seg | AP-seg | PRO-seg |
|---------|--------|---------|-----------|--------|--------|-----------|--------|--------|---------|
| MVTec-CA | PatchCore | 4-shot | 91.2 | 91.9 | 97.1 | 96.0 | 67.7 | 68.2 | 87.2 |
| | WinCLIP | 4-shot | 94.4 | 93.3 | 97.6 | 95.0 | 63.3 | - | 87.0 |
| | APRIL-GAN | 4-shot | 83.9 | 87.8 | 93.8 | 93.5 | 54.7 | 49.0 | 90.8 |
| | GraphCore | 4-shot | 93.2 | - | - | 97.5 | - | - | - |
| | CoDeGraph (Ours) | 0-shot | 98.5 | 97.8 | 99.6 | 98.1 | 73.8 | 77.2 | 95.4 |
| | PatchCore | full-shot | 98.6 | - | - | 97.9 | - | - | 91.9 |
| | CFlow | full-shot | 98.4 | - | - | 98.4 | - | - | 93.5 |
| | DDAD | full-shot | 99.1 | - | - | 98.1 | - | - | 92.8 |

Table 11: Quantitative comparisons on the **Inconsistent-Anomaly Dataset**. We compared CoDeGraph with few-shot and full-shot methods. Bold indicates the best performance. All metrics are in %.

| Dataset | Method | Setting | AUROC-cls | F1-cls | AP-cls | AUROC-seg | F1-seg | AP-seg | PRO-seg |
|---------|--------|---------|-----------|--------|--------|-----------|--------|--------|---------|
| MVTec-IA | PatchCore | 4-shot | 90.3 | 94.7 | 95.3 | 94.1 | 47.5 | 44.3 | 83.0 |
| | WinCLIP | 4-shot | 95.4 | 95.1 | 97.2 | 96.5 | 58.6 | - | 89.5 |
| | APRIL-GAN | 4-shot | 95.0 | 94.0 | 96.9 | 96.5 | 57.4 | 55.9 | 92.1 |
| | GraphCore | 4-shot | 92.8 | - | - | 97.4 | - | - | - |
| | CoDeGraph (Ours) | 0-shot | 98.3 | 97.3 | 99.1 | 98.2 | 65.1 | 65.8 | 94.4 |
| | PatchCore | full-shot | 99.1 | - | - | 98.0 | - | - | 93.4 |
| | CFlow | full-shot | 98.3 | - | - | 98.7 | - | - | 94.9 |
| | DDAD | full-shot | 99.6 | - | - | 98.1 | - | - | 92.9 |

# C Implementation Details

## C.1 Construction of the MVTec-SynCA Dataset

To create MVTec-SynCA, for each object subclass, we randomly selected a representative anomalous image and applied a series of geometric and photometric transformations to simulate real-world imaging variations. The applied transformations included:

- Randomly applied rotations of $\pm 15°$.

- Randomly applied translations of $\pm 2.5\%$ of image dimensions in both x and y directions.

- Randomly applied brightness, contrast and saturation adjustments of $\pm 10\%$ to 80% of generated images.

- Applied Gaussian noise ($\sigma = 7.5$) and salt-and-pepper noise (1% of total pixels) to 20% of the generated images.

All transformations were applied to both the anomalous images and their corresponding ground truth masks using reflection padding to maintain boundary integrity. For each subclass, the number of synthetic anomalies added to each subclass is approximately 15% of the total images. For the consistent-anomaly subclass `metal_nut`, we kept the original dataset unchanged as the number of consistent-anomaly images already exceeds 20% of the total images.

## C.2 Construction of the ConsistAD Dataset

The ConsistAD dataset was designed as a comprehensive benchmark for evaluating consistent anomaly detection by aggregating test images from three established anomaly detection datasets: MVTec AD, MVTec LOCO, and MANTA. It totally comprises 9 subclasses: `cable`, `metal_nut`, and `pill` from MVTec AD; `breakfast_box` and `pushpins` from MVTec LOCO; and `capsule`, `coated_tablet`, `coffee_bean`, and `red_tablet` from MANTA. The dataset construction process for each source is detailed below:

- **MVTec AD and MVTec LOCO**: We directly used the original test images from the specified subclasses without modifications.

- **MANTA**: The MANTA dataset consists of multi-view images, where each image concatenates five views of an object captured from different viewpoints, tailored for multi-view anomaly detection. To adapt these for our purpose, we extracted single-view images as follows:
  - For normal images, we selected the middle view.
  - For anomalous images, we chose the view with the largest anomaly mask area, as some views may not display anomalies.
  - From each subclass, we randomly sampled 100 normal images and 50 anomalous images. To emphasize consistent anomalies, which often exhibit large anomaly masks in MANTA, we ensured that 40% of the anomalous images (20 images) were those with the largest anomaly masks, while the remaining 40 were randomly selected from the other anomalous views.

## C.3 Detailed Results

This section provides detailed per-class results for CoDeGraph on individual datasets, showing the performance breakdown across all object categories.

Table 12: Detailed per-class results for CoDeGraph on MVTec AD dataset. All metrics are in %.

| Class | AUROC-cls | F1-cls | AP-cls | AUROC-seg | F1-seg | AP-seg | PRO-seg |
|---|---|---|---|---|---|---|---|
| bottle | 100.00 | 100.00 | 100.00 | 98.68 | 79.98 | 83.18 | 96.25 |
| cable | 99.44 | 97.80 | 99.69 | 98.30 | 69.25 | 72.35 | 91.97 |
| capsule | 96.29 | 95.54 | 99.20 | 99.11 | 53.84 | 53.30 | 97.41 |
| carpet | 99.92 | 99.44 | 99.97 | 99.39 | 73.73 | 75.18 | 97.43 |
| grid | 98.50 | 95.65 | 99.48 | 98.34 | 47.16 | 41.75 | 94.96 |
| hazelnut | 99.04 | 97.22 | 99.40 | 99.30 | 71.63 | 71.66 | 95.44 |
| leather | 100.00 | 100.00 | 100.00 | 99.69 | 62.04 | 65.77 | 98.30 |
| metal_nut | 99.85 | 99.46 | 99.97 | 98.03 | 83.50 | 85.87 | 96.28 |
| pill | 96.07 | 96.22 | 99.27 | 97.86 | 68.72 | 73.26 | 97.85 |
| screw | 91.04 | 92.19 | 95.00 | 99.12 | 49.24 | 45.74 | 96.09 |
| tile | 100.00 | 100.00 | 100.00 | 97.99 | 76.49 | 79.06 | 95.56 |
| toothbrush | 100.00 | 100.00 | 100.00 | 99.59 | 74.47 | 75.69 | 96.43 |
| transistor | 98.46 | 92.31 | 97.87 | 91.84 | 60.55 | 60.54 | 77.14 |
| wood | 96.49 | 95.87 | 98.83 | 97.46 | 68.81 | 74.49 | 93.38 |
| zipper | 99.76 | 99.17 | 99.94 | 98.36 | 62.82 | 63.09 | 94.35 |
| **Mean** | **98.32** | **97.39** | **99.24** | **98.20** | **66.82** | **68.06** | **94.59** |

Table 13: Detailed per-class results for CoDeGraph on Visa dataset. All metrics are in %.

| Class | AUROC-cls | F1-cls | AP-cls | AUROC-seg | F1-seg | AP-seg | PRO-seg |
|---|---|---|---|---|---|---|---|
| candle | 95.44 | 88.32 | 95.63 | 99.36 | 40.69 | 32.64 | 95.62 |
| capsules | 88.15 | 84.82 | 93.76 | 98.67 | 51.03 | 43.61 | 85.55 |
| cashew | 95.52 | 93.68 | 98.03 | 99.28 | 75.37 | 79.10 | 91.87 |
| chewinggum | 98.24 | 95.92 | 99.22 | 99.33 | 52.46 | 50.67 | 92.60 |
| fryum | 97.52 | 94.79 | 98.93 | 98.04 | 59.26 | 52.53 | 87.37 |
| macaroni1 | 87.97 | 82.35 | 87.73 | 99.48 | 19.82 | 12.40 | 95.68 |
| macaroni2 | 57.17 | 68.53 | 54.13 | 96.61 | 6.85 | 1.86 | 83.16 |
| pcb1 | 91.57 | 86.38 | 90.47 | 99.57 | 79.62 | 87.79 | 94.74 |
| pcb2 | 93.84 | 91.37 | 94.43 | 97.74 | 35.66 | 24.15 | 88.10 |
| pcb3 | 95.75 | 90.38 | 96.01 | 98.42 | 40.85 | 42.61 | 92.43 |
| pcb4 | 99.42 | 96.12 | 99.42 | 98.88 | 48.39 | 45.40 | 92.80 |
| pipe_fryum | 98.52 | 95.65 | 99.24 | 99.45 | 69.08 | 71.82 | 96.78 |
| **Mean** | **91.59** | **89.03** | **92.25** | **98.73** | **48.26** | **45.38** | **91.39** |

Table 14: Detailed per-class results for CoDeGraph on MVTec-SynCA dataset. All metrics are in %.

| Class | AUROC-cls | F1-cls | AP-cls | AUROC-seg | F1-seg | AP-seg | PRO-seg |
|---|---|---|---|---|---|---|---|
| bottle | 100.00 | 100.00 | 100.00 | 98.75 | 79.64 | 83.11 | 96.49 |
| cable | 99.56 | 98.71 | 99.81 | 97.84 | 65.53 | 66.58 | 92.12 |
| capsule | 90.51 | 93.02 | 98.34 | 95.18 | 37.64 | 32.76 | 92.23 |
| carpet | 99.57 | 99.54 | 99.89 | 99.00 | 71.04 | 71.78 | 94.06 |
| grid | 98.84 | 97.14 | 99.67 | 97.28 | 42.66 | 36.28 | 86.01 |
| hazelnut | 99.02 | 98.34 | 99.49 | 98.98 | 66.26 | 60.07 | 96.10 |
| leather | 100.00 | 100.00 | 100.00 | 99.59 | 58.25 | 68.42 | 97.65 |
| metal_nut | 99.85 | 99.46 | 99.97 | 98.03 | 83.50 | 85.87 | 96.28 |
| pill | 96.09 | 96.55 | 99.39 | 97.46 | 74.69 | 79.69 | 97.16 |
| screw | 73.39 | 87.76 | 90.61 | 98.07 | 46.32 | 40.82 | 89.50 |
| tile | 98.75 | 97.06 | 99.63 | 93.28 | 67.73 | 67.97 | 82.30 |
| toothbrush | 100.00 | 100.00 | 100.00 | 99.51 | 81.42 | 83.28 | 94.27 |
| transistor | 99.04 | 94.64 | 99.03 | 90.14 | 54.55 | 52.63 | 70.62 |
| wood | 97.33 | 96.60 | 99.28 | 97.50 | 70.58 | 77.39 | 91.92 |
| zipper | 99.66 | 99.31 | 99.92 | 97.13 | 48.20 | 42.18 | 89.55 |
| **Mean** | **96.78** | **97.21** | **99.00** | **97.18** | **63.20** | **63.25** | **91.09** |

Table 15: Detailed per-class results for CoDeGraph on ConsistAD dataset. All metrics are in %.

| Class | AUROC-cls | F1-cls | AP-cls | AUROC-seg | F1-seg | AP-seg | PRO-seg |
|---|---|---|---|---|---|---|---|
| breakfast_box | 76.99 | 69.57 | 77.89 | 85.08 | 41.59 | 44.48 | 63.10 |
| cable | 99.44 | 97.80 | 99.69 | 98.30 | 69.25 | 72.35 | 91.97 |
| capsule | 94.68 | 91.30 | 94.06 | 92.46 | 69.39 | 72.77 | 85.38 |
| coated_tablet | 99.58 | 98.99 | 99.41 | 99.53 | 82.01 | 91.03 | 98.10 |
| coffee_bean | 91.58 | 82.35 | 86.35 | 73.17 | 26.46 | 24.61 | 68.48 |
| metal_nut | 99.85 | 99.46 | 99.97 | 98.03 | 83.50 | 85.87 | 96.28 |
| pill | 96.07 | 96.22 | 99.27 | 97.86 | 68.72 | 73.26 | 97.85 |
| pushpins | 61.79 | 59.09 | 57.70 | 52.57 | 11.08 | 6.95 | 56.60 |
| red_tablet | 99.34 | 96.08 | 98.68 | 85.00 | 51.19 | 45.93 | 85.12 |
| **Mean** | **91.04** | **87.87** | **90.33** | **86.89** | **55.91** | **57.47** | **82.54** |

