# OpenReview forum: "On the Problem of Consistent Anomalies in Zero-Shot Industrial Anomaly Detection"
_TMLR — Accepted by TMLR_

### Review · Reviewer_yJpR · 2025-08-02

**Summary Of Contributions:**

This paper introduces CoDeGraph, a novel zero-shot anomaly detection (AC) and segmentation (AS) framework tailored to address the overlooked challenge of consistent anomalies in industrial inspection datasets. Unlike existing methods that suffer performance drops when anomalies recur across images, CoDeGraph exploits a new insight called the **neighbor-burnout phenomenon**, in which consistent anomalies exhibit abrupt drops in similarity growth that is quantified via an **endurance ratio**. This metric is used to construct an image-level graph whose dense communities correspond to consistent anomalies. By identifying and filtering out high-dependency patches from these communities, CoDeGraph significantly improves performance. Theoretical justification is provided via Extreme Value Theory, and extensive benchmarks on MVTec AD, Visa, and newly proposed datasets MVTec-CA, MVTec-SynCA, ConsistAD show improved zero-shot performance in classification and segmentation tasks.

**Strengths**:
1. Well-motivated and novel identification of consistent anomaly issue
2. Clear formalization of "neighbor-burnout" and strong theoretical backing in Section 3.2
3. Impressive experimental improvements (up to +14.9% F1, +18.8% AP) across 3 datasets and cls. and seg. tasks
4. Robustness validated across feature extractors (ViT, DINOv2) and other frameworks (e.g., AnomalyCLIP, WinCLIP. etc)
5. Thorough ablation studies and practical considerations (inference time, scalability).

**Weaknesses**:
1. Despite claiming zero-shot capability and conducting zero-shot experiments, the method makes extensive assumptions about the structure of test data (e.g., normal patterns outnumber anomalies, consistent anomalies form dense communities) that may not hold in real-world deployments
2. Relies on heuristic thresholding (e.g., 80th percentile of normal patch scores) to define consistent anomalies, introducing circularity in performance evaluation and may not be robust across all datasets.
3. The method’s performance is sensitive to hyperparameters such as ω (burnout threshold) and α (weighting exponent), which are currently manually tuned. This tuning undermines claims of zero-shot generality. Some hyperparameters (e.g., ω, α) could be further auto-tuned or learned through parameter aware training.
4. The paper does not quantify or visualize failure cases (e.g., over-filtering of legitimate patterns), especially in complex test distributions like MVTec LOCO.
5. Assumes access to the full test set at inference to build graphs and communities, which is incompatible in streaming and online detection cases

**Audience:**

Yes

**Audience Explanation:**

The paper introduces a novel problem setting around consistent anomalies in zero-shot segmentation. The authors propose an elegant and generally applicable graph-based solution. Researchers in industrial vision, unsupervised representation learning, and zero-shot transfer would find the method and insights (e.g., neighbor-burnout) both practically and theoretically valuable.

The work also exemplifies how one can blend statistical modeling (EVT) with modern ViT-based features for structured anomaly detection, which should interest the broader TMLR community.

**Broader Impact Concerns:**

No major ethical concerns. Authors addressed the concern of deploying such systems in critical applications in "Broader Impact Statement".

**Claims And Evidence:**

Yes

**Claims Explanation:**

- The empirical results on consistent-anomaly benchmarks are strong and clearly support the core claims. For instance, in Table 1, CoDeGraph improves the pixel-wise F1 score by +14.9% and AP by +18.8% over the strongest prior zero-shot method (MuSc 10%) on MVTec-CA, and similar gains are observed on MVTec-SynCA and ConsistAD. These results substantiate the claim that existing methods fail under consistent anomaly conditions and that CoDeGraph mitigates this issue effectively

- The neighbor-burnout phenomenon is supported both theoretically and empirically. The authors derive growth dynamics from Extreme Value Theory (Appendix A.1), showing that:
“τ(i)(x) follows an exponential distribution with rate proportional to i, under certain assumptions such as distances from a normal patch x to other normal patches having an asymptotic power law tail ” (Page 4)
They validate this in Figure 2, where growth rates for consistent-anomaly patches (orange) exhibit sharp spikes, distinguishing them from the smooth decay in normal or inconsistent anomalies. The endurance ratio ζ(x, I(i)) is derived from this and further supported by Figure 3(a).

**Requested Changes:**

1. Evaluation circularity via ground truth thresholds (Section 4.1 Evaluation Metrics):
The paper defines consistent anomalies based on whether their anomaly scores fall below the 80th percentile of normal patch scores, using ground-truth labels. This creates a circular dependency between how anomalies are defined and how performance is evaluated, which may artificially favor the proposed method.
To strengthen the evaluation, the authors should either remove reliance on ground truth when selecting consistent anomalies or include an ablation showing how performance varies with different (a comparison between 2 values suffices) unsupervised thresholds. This would improve confidence in the general applicability of the evaluation protocol.

2. Missing failure mode analysis:
The manuscript acknowledges in Section 6 Limitations that the method may underperform in certain cases (e.g., when anomaly images outnumber normal ones), but provides no concrete examples. Including both qualitative and quantitative failure cases, such as incorrect filtering in multi-modal normal categories or poor performance when consistent anomalies dominate, would clarify the framework's limitations.

3. Hyperparameter sensitivity not fully addressed (Section 4.3):
Key parameters such as α (weighting for endurance ratio) and ω (burnout index) are manually tuned per dataset, which contradicts the method's zero-shot claims. The authors should report results using fixed or cross-dataset settings, or provide performance ranges under moderate variation. This would demonstrate robustness without requiring dataset-specific calibration.
Alternatively, the authors could consider evaluating the method across a broader set of consistent-anomaly (CA) datasets with similar structural patterns, which would help confirm the generalizability of the chosen hyperparameter settings.

4. Theoretical assumptions behind neighbor-burnout (Section 3.2, Appendix A.1):
The modeling of neighbor-burnout using EVT assumes that patch-to-patch distances are i.i.d. and follow a power-law distribution in the tail. However, features from Vision Transformers are known to be highly structured and spatially correlated, especially in industrial images where defects may align with visual textures or lighting patterns. Prior work (e.g., Athalye et al., 2018) has shown that deep network feature spaces violate i.i.d. assumptions, calling into question the applicability of classical EVT. The authors are encouraged to empirically verify these assumptions or qualify the limitations of their theoretical model accordingly. This would help clarify the scope and robustness of the endurance ratio’s theoretical foundation.
Athalye, A., Engstrom, L., Ilyas, A., & Kwok, K. (2018). Synthesizing Robust Adversarial Examples. Proceedings of ICML 2018. https://arxiv.org/abs/1707.07397

5. Optional: Incomplete runtime and memory profiling (Table 5):
While inference time is reported, memory usage and the cost of graph construction and community detection are not detailed. Including GPU memory and per-stage latency would help assess scalability, especially for large test sets.

---

> ### Author Response · Authors · 2025-09-04
>
> We sincerely appreciate your valuable feedback and provide the following detailed responses to all weaknesses and requested changes.
>
> ---
>
> ### Weaknesses
>
> 1. **Assumptions about test data (normal outnumber anomalies; consistent anomalies form dense communities):**
>    We clarify that our method does not assume "consistent anomalies form dense communities". Indeed, dense communities emerges as consequence of the graph construction and neighbor-burnout phenomenon (Sections 3.1–3.3).
>    The assumption that normal patterns outnumber anomalies is standard and realistic in industrial anomaly detection, supported by benchmarks such as MVTec AD (>95% normal pixels) and Visa (>98%). Furthermore, none of the open-source datasets we knew contain >20% consistent-anomaly samples, reflecting real-world quality control scenarios where defects are rare.
>
> 2. **Heuristic thresholding (80th percentile) introduces circularity and lacks robustness:**
>    The 80th percentile threshold is used only post-hoc to define consistent anomalies for evaluation, aligning with Definition 3.2, and is essential for benchmarking datasets without clear consistent-anomaly patterns (e.g., Visa, MVTec-IA). CoDeGraph remains fully zero-shot and unsupervised: labels are never used during inference, only for evaluation metrics (AUROC, F1-max, Capture Rate, ...), consistent with prior work. We will clarify this in Section 4.1:
>    *“This threshold uses ground-truth solely for post-hoc analysis and evaluation metrics, ensuring no impact on the zero-shot nature of
> CoDeGraph.”*
>
> 3. **Hyperparameter sensitivity and manual tuning hyperparameter per dataset:**
>    We clarify that all main experiments use fixed hyperparameters across datasets ($\omega=0.3N, \alpha=0.2, K=0.1N, \tau=0.95, r\in\{1,3,5\}, k_{\text{IQR}}=4.5$), with no per-dataset tuning. Only in ablation study, we experimented with the hyper-parameters, and the results (Table 6, Figure 7; Appendix B.3, B.7) confirm robustness of our method. We will clarify this in Implementation Details:
>    *“All experiments use fixed parameters (described below) across all datasets, unless specified in ablations”*
>
> 4. **Lack of failure mode analysis (e.g., over-filtering in MVTec LOCO):**
>    We have expanded our discussion on failure modes in Appendix B.2 (Multi-Modal Industrial Objects and Failure Modes), focusing on cases where consistent anomalies outnumber specific normal variants in multi-modal settings. A controlled experiment using MVTec LOCO data (8 tomato, 4 orange, and 4 banana juice bottles as normal variants, with 8 orange bottles lacking labels as logical anomalies) demonstrates that less frequent normal variants (orange, banana) may form dense clusters, risking misclassification as anomalous communities. This highlights a limitation in rare scenarios where consistent anomalies dominate, though such cases are uncommon (<20% consistent-anomaly samples in evaluated datasets). We will strengthen the cross-reference in Section 6 to guide readers to Appendix B.2 for these insights.
>
> 5. **Dependence on full test set, limiting streaming/online applicability:**
>    Like prior zero-shot baselines (e.g., MuSc), CoDeGraph is designed for batch zero-shot industrial anomaly detection, as discussed in [1]. We do not claim streaming capability of CoDeGraph but it supports smaller batch processing (e.g., 21–84 images per batch for MVTec AD, enabled by random subset division), demonstrating applicability in scenarios without access to the full dataset.
>
> ---
>
> ### Requested Changes
>
> 1. **Evaluation circularity (Section 4.1):** See Weakness 2.
> 2. **Failure mode analysis:** See Weakness 4.
> 3. **Hyperparameter sensitivity (Section 4.3):** See Weakness 3.
> 4. **Theoretical assumptions of neighbor-burnout (i.i.d. patches):**
>    We acknowledged that Vision Transformer (ViT) features exhibit spatial correlations (Appendix A.1.3) challenge the i.i.d. assumption in our EVT model for similarity growth dynamics (Theorem 1). However, the large number of patches per image (M=1369) justifies this approximation, as it enables mathematical tractability while capturing the statistical behavior of distances. This is also empirically supported by the predictions on the distribution of endurance ratio: $ Y_{(i)}/Y_{(\omega)} $ follows a Beta distribution with $\beta \approx 1$. For your reference, we analyzed the correlation matrix of patch-to-patch distance vectors $ X_{p_1}, \ldots, X_{p_{1369}} $ in the MVTec AD cable subclass, finding that over 50% of absolute pairwise correlations $ |corr(X_{p_i}, X_{p_j})| < 0.2 $. While not proving i.i.d., this suggests weak dependencies, supporting the reasonability of our assumption.
>
> 5. **Runtime/memory profiling (Table 5):** Appendix B.4 details computational and memory considerations, omitted from the main text due to space limits.
>
> ---
> References:
> [1] Damm, Simon, et al. *Anomalydino: Boosting patch-based few-shot anomaly detection with dinov2.* WACV 2025.

---

### Review · Reviewer_MfYc · 2025-08-05

**Summary Of Contributions:**

Current anomaly detection methods can fail when “consistent anomalies” are present in the test set, such as when a machine produces similar defects for multiple objects. This failure can be attributed to a lack of anomaly signal when considering only the most similar image. This is addressed by considering the distribution of similarities across all images, providing a clearer change in distribution between anomalous and in-distribution imagery. The endurance ratio enables this change in distribution to be characterized and a graph of suspicious links to be constructed. Communities of anomalous image patches are detected in the graph and removed from consideration for a standard novelty detection procedure. Experiments show that the proposed CoDeGraph method better addresses consistent anomalies while maintaining parity with other methods on standard benchmarks.

**Audience:**

Yes

**Audience Explanation:**

The issue of consistent anomalies proposed in the paper is compelling and the high-level solution to the problem intuitive. The proposed method neatly resolves the implementation challenges of the high-level solution.

**Broader Impact Concerns:**

None.

**Claims And Evidence:**

Yes

**Claims Explanation:**

The figures throughout Section 3 do a great job of providing intuition for the individual concepts introduced in this paper. I especially like how Figure 2B shows a limitation of the method: there is little/no benefit when consistent anomalies are not present.

The experiments are very thorough, covering data with and without consistent anomalies, segmentation maps, and various ablations.

However, I think the paper needs to provide more guidance on what the full CoDeGraph method looks like. The paper introduces a lot of bespoke concepts, terminology, and algorithms, so working out how they connect was challenging to me. I think that a figure showing these connections would address this issue.

There’s an incongruity between the EVT model being mentioned as part of a core contribution, then relegated to the Appendix with little discussion in the core paper.

**Requested Changes:**

C1: I think the equation following Definition 3.2 only makes sense if $H < K$, but this condition is not stated. Please add this condition or explain whether the equation holds when the condition is violated.

C2: Provide more figure and/or text guidance for what the full CoDeGraph method looks like.

C3: Remove the “Setting” column of Tables 1-2 and use the extra space to increase the font size.

C4: Either improve the EVT discussion in the main paper or remove it from the first contribution.

---

> ### Author Response · Authors · 2025-09-04
>
> We sincerely appreciate your valuable feedback and provide the following detailed responses to all weaknesses and requested changes.
>
> ### Requested Changes
>
> **C1: The equation following Definition 3.2 only makes sense if H <K, but this condition is not stated.**
> Thank you for noting this. We have revised the equation in Section 3.2 (page 10) to use $ H_0 = \min(H, K) $ instead of H. This ensures the bound mathematically correct.
>
> **C2: Provide more guidance for what the full CoDeGraph method looks like.**
> We added a diagram for CoDeGraph pipeline in Section3.1 (Figure 2).
>
> **C3: Remove the “Setting” column of Tables 1-2 and use the extra space to increase the font size.**
> We removed the “Setting” column from Tables 1 and 2.
>
> **C4: Improve the EVT discussion in the main paper or remove it from the first contribution.**
> We value your note about the EVT model’s placement. In the revision, we expanded the discussion on EVT in Section 3.2.

---

### Review · Reviewer_DTjc · 2025-09-03

**Summary Of Contributions:**

The paper introduces CoDeGraph, a zero-shot industrial anomaly detection method tailored to “consistent anomalies” that recur across images. It proposes an image-level graph representation with community detection to exploit cross-image consistency for robust anomaly localization and classification. The authors identify and formalize a “neighbor-burnout” phenomenon—normal patches retain many neighbors while anomalous patches quickly exhaust them—and provide an Extreme Value Theory-based explanation for its discriminative power. Extensive experiments show state-of-the-art performance on standard industrial benchmarks with strong backbone-agnostic robustness.

**Audience:**

Yes

**Audience Explanation:**

The paper is about image anomaly classification and anomaly segmentation, which will attract TMLR audience in computer vision and image processing.

**Claims And Evidence:**

Yes

**Claims Explanation:**

The claims are supported by theoretical grounding and solid empirical studies.

**Requested Changes:**

Strengths:
- Strong and timely problem framing: clearly identifies “consistent anomalies” as a failure mode for zero-/few-shot industrial inspection and motivates why standard nearest-neighbor or feature-matching pipelines struggle.
- Methodologically sound design: constructs an image-level graph (images as nodes, edges via cross-image similarity of anomaly patterns) and leverages community detection to isolate and down-weight consistent anomalies; the pipeline remains training-free and backbone-agnostic.
- Theoretical grounding: provides an Extreme Value Theory (EVT) perspective that explains why “neighbor-burnout” is discriminative, offering a principled underpinning rather than a purely empirical heuristic.
- Solid empirical evidence on dedicated benchmarks (e.g., MVTec-CA, MVTec-SynCA, ConsistAD) demonstrating notable gains on the targeted consistent anomaly regime.
- Clear writing and presentation: motivation, intuition, and empirical setup are generally easy to follow.

Weaknesses:
- While this method demonstrates clear advantages over existing approaches on benchmarks containing only consistent anomalies, the evaluation would be more convincing if it also included benchmarks with both consistent and inconsistent anomalies (e.g., the full MVTec AD dataset). Since the paper is positioned as zero-shot industrial inspection, it is important to validate performance in scenarios without prior knowledge of anomaly types, which better reflects real-world applications.
- The paper is generally well-written. However, it would be helpful to include some figures that illustrate key aspects of the algorithm, as this would improve clarity and enhance the reader’s understanding.

Requested changes:
- Add a pipeline diagram.
- Add experiments on full MVTec AD.

---

> ### Author Response · Authors · 2025-09-04
>
> We sincerely appreciate your valuable feedback and provide the following detailed responses to all weaknesses and requested changes.
>
> ## Weaknesses
> **1: Lack of experiments on general industrial datasets**
> We clarify that we have already conducted experiments on the full MVTec AD dataset (MVTec AD = MVTec-CA + MVTec IA). Detailed per-class results are provided in Appendix C.3, Table 12. Additionally, we performed experiments on the Visa dataset (Table 2, Table 13) confirm CoDeGraph’s performance on general industrial objects, not just datasets with consistent-anomaly patterns.
>
> **2: No pipeline diagram**
> We added a diagram for CoDeGraph pipeline in Section3.1 (Figure 2).
>
> ## Requested changes:
> **1: Add experiments on full MVTec AD**
> See Weakness 1.
>
> **2: Add a pipeline diagram**
> See Weakness 2.

---

> > ### Comment · Reviewer_DTjc · 2025-09-22
> > **Thank you for the response**
> >
> > Thank you to the authors for providing the clear diagram of the CoDeGraph pipeline. Regarding the experiments on the full MVTec AD dataset, I see that detailed per-class results for CoDeGraph are already reported in Table 12. My main concern is the comparison with existing methods on this dataset. Does CoDeGraph also demonstrate clear advantages over previous approaches?

---

> ### Author Response · Authors · 2025-09-22
> **Response to reviewer DTjc question about full MVTec benchmark.**
>
> Dear Reviewer DTjc,
>
> Thank you for your feedback and for noting the addition of the CoDeGraph pipeline diagram. Regarding your question about CoDeGraph’s performance on the full MVTec AD dataset, we clarify that **MVTec AD = MVTec-CA (metal_nut, cable, pill) + MVTec-IA (all other subclasses)**. Hence, Our method achieves **state-of-the-art (SOTA) performance on MVTec-AD** by outperforming all prior approaches on MVTec-CA while remaining competitive on MVTec-IA. This results in overall better performance on the full MVTec AD benchmark, as shown in Table below (we can get this by combine results from Table 1 and Table 2 in the paper):
>
> | **Method**      | **AUROC-cls** | **F1-cls** | **AP-cls** | **AUROC-seg** | **F1-seg** | **AP-seg** |
> |-----------------|---------------|------------|------------|---------------|------------|------------|
> | **Ours**        | **98.3**      | 97.4       | **99.2**   | **98.2**      | **66.8**   | **68.1**   |
> | MuSc            | 97.8          | **97.5**   | 99.1       | 97.3          | 62.6       | 62.7       |
> | ACR             | 85.8          | 91.3       | 92.9       | 92.5          | 44.2       | 38.9       |
> | APRIL-GAN       | 86.1          | 90.4       | 93.5       | 87.6          | 43.3       | 40.8       |
> | AnomalyCLIP     | 91.5          | 92.7       | 96.3       | 91.1          | 39.1       | 34.5       |
>
> > **Note**: Bold values indicate the best performance per metric. CoDeGraph achieves the highest score in **five of six metrics** (all except F1-cls, where it is competitive), confirming its effectiveness for zero-shot anomaly detection on MVTec AD.
>
> Additionally, with the DINOv2 backbone (Table 9 in the revision), CoDeGraph attains **69.1%  F1-seg** (+6.5% over MuSc) and **71.9% AP-seg** (+9.2% over MuSc), representing the current SOTA for zero-shot/few-shot anomaly segmentation on MVTec AD (to the best of our knowledge).
>
> We are happy to provide further details upon request.
>
> Sincerely,
> The Authors

---

> > ### Comment · Reviewer_DTjc · 2025-09-23
> >
> > Thanks for providing the additional comparison on the full MVTec AD dataset. From the results, the proposed method achieves substantially better performance than prior approaches in terms of F1-seg and AP-seg, and slightly better results on the remaining metrics, which are already near-saturated.
> >
> > These results satisfactorily address my earlier concerns.

---

> > > ### Author Response · Authors · 2025-09-23
> > >
> > > Thank you for your thorough review and positive feedback. We’re glad that our response addressed your concerns. We appreciate your comments, which strengthened our work, and look forward to your official recommendation.
> > >
> > > Sincerely,
> > >
> > > The Authors

---

### Author Response · Authors · 2025-09-04
**Global response to reviewers.**

Dear Editor and Reviewers,

Thank you for your valuable feedback on our manuscript. We have incorporated your suggestions in our revision, with revised text highlighted in blue. To summarize, besides minor clarifications and typo corrections, we made the following revisions:

- We added a diagram illustrating the CoDeGraph pipeline in Section 3.1 to enhance clarity of the workflow.
- We corrected the equation for the anomaly score bound following Definition 3.2.
- We extended the discussion on the Extreme Value Theory (EVT) model in Section 3.2.
- We added a failure mode analysis in Appendix B.2, including qualitative and quantitative examples to clarify limitations.
- We clarified implementation details and evaluation metrics throughout to improve transparency and reproducibility.
- We change the title of the paper to "On the Problem of Consistent Anomalies in Zero-Shot Industrial Anomaly Detection"

We are happy to provide further clarifications or revisions if needed.

Best,
The Authors

---

### Decision · Action_Editor_nTv5 · 2025-10-10

**Recommendation:** Accept as is

**Additional Comments:**

The reviewers highly appreciated the paper's novel problem setting of "consistent anomaly."
They also found the experiments effectively demonstrated the advantages of the proposed method.
The reviewers suggested a few updates concerning the experimental setup details and the theoretical aspects, which the authors have addressed appropriately.
Overall, based on the novelty of the proposed problem, the solid methodology, and the strong experimental evidence, the reviewers recommend acceptance of this paper.

**Audience:**

Yes

**Audience Explanation:**

Anomaly detection is a major research topic in machine learning and is of interest to TMLR readers.

**Claims And Evidence:**

Yes

**Claims Explanation:**

This paper proposes a novel anomaly detection technique for "consistent anomalies," which are similar defects that consistently appear across multiple images
The proposed method first computes distances between image patches using the intermediate representations of a ViT.
This distance is then used to construct a graph that represents the similarity between images.
The paper reported that consistent anomalies exhibit distinctive behavior within this graph.
Based on this, the authors proposed an anomaly detection method using community detection to identify the consistent anomalies.
They demonstrated the effectiveness of the proposed method using various variations of the MVTec dataset.

---

> ### Author Response · Authors · 2025-10-14
>
> Once again, we appreciate the reviewers and the Action Editor for their careful effort in evaluating our paper.